# SymTex: A New Benchmark for Non-monotonic Reasoning Capability of Large Language Models

## Abstract

Non-monotonic reasoning (NMR) plays a crucial role in logical reasoning, allowing inference to adjust as new information arises. This adaptability is key for large language models (LLMs) to handle complex problems and adjust reasoning in dynamic environments, mimicking human-like flexibility in thought. Recent works mainly explore using LLMs to address non-monotonic reasoning through textual logic representation, as LLMs excel in understanding natural language. However, textual logic representation often leads to ambiguity and complexity, especially in complex situations, while symbolic logic representation is more clear and precise, avoiding these issues. In this work, we introduce a framework called Multi-step Generation for Symbolic and Textual NMR Samples (MG-SymTex) to generate diverse non-monotonic samples automatically, and build a non-monotonic reasoning benchmark, called SymTex, which is used to evaluate the non-monotonic reasoning capability of LLMs. SymTex comprises two types of description and three types of predicate, facilitating two primary tasks: Tri-State Boolean Querying and Answer Set Computation. Through our comprehensive evaluations, we demonstrate that state-of-the-art LLMs such as *gpt-4o*, *claude-3.5-sonnet*, and *o1-mini* encounter significant challenges when addressing our proposed benchmark, highlighting the difficulty of non-monotonic reasoning in LLMs.

## 1 Introduction

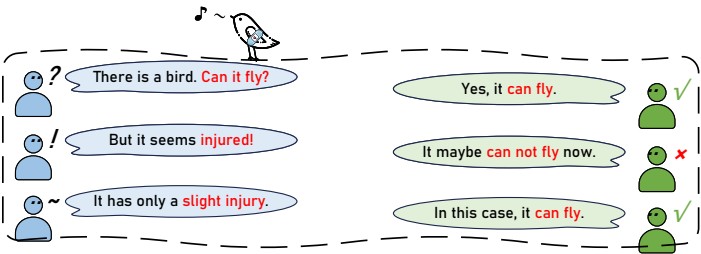

Figure 1: An example of non-monotonic reasoning in daily life.

Non-monotonic reasoning (NMR) is a complex and essential component of logical reasoning. Compared with monotonic reasoning, it introduces the ability to invalidate previously established conclusions when additional information is incorporated (Ginsberg, 1980; Reiter, 1988). Thus, it enables a more adaptive and context-sensitive inference process. Enhancing non-monotonic reasoning ability allows LLMs to adapt their reasoning pathways when confronted with conflicting or ambiguous information, ensuring that LLMs can maintain reasonable reasoning and judgment in the face of evolving information. McDermott & Doyle (1980) demonstrated that non-monotonic logics can invalidate previously established theorems by introducing new axioms. Some key formalisms within non-monotonic reasoning include Default Reasoning (Reiter, 1980), Abductive Inference (Josephson & Josephson, 1996), and Belief Revision (Darwiche & Pearl, 1997).

In routine decision-making, non-monotonic reasoning closely mirrors human cognitive processes, where initial decisions are often based on habitual or default assumptions, and later adjusted as new,

Table 1: The difference between SymTex and others. "Commonsense-driven" indicates the need for commonsense knowledge in reasoning tasks. Abbreviations in the operations column represent supported logical operations: SN (Strong Negation), DN (Default Negation), Disj (Disjunction), and Cons (Constraint). Predicate style refers to the format of predicates: RandS (Random String), RandW (Random Word), and RelW (Related Word). The Multi-ary predicate and Multi-objects columns specify the maximum number of arguments and objects involved in tasks, with N/A indicating no specification.

| Dataset | Multi-ary Predicate | Multi Objects | Commonsense-Driven | Fact-rule-query | Non-monotonic | Operations | Logic Style | Predicate Style |
|---|---|---|---|---|---|---|---|---|
| $\delta$-NLI (Rudinger et al., 2020) | | | $\checkmark$ | $\checkmark$ | $\checkmark$ | | textual | RelW |
| ProofWriter (Tafjord et al., 2021) | 2 | 2 | $\times$ | $\checkmark$ | $\times$ | SN | textual | RandW |
| ruletaker (Clark et al., 2021) | | | $\times$ | $\checkmark$ | $\times$ | SN | textual | RandW |
| LogicNMR (Xiu et al., 2022) | 1 | 1 | $\times$ | $\checkmark$ | $\checkmark$ | SN,DN | textual | RandW |
| generics-exemplars (Allaway et al., 2023) | | | $\checkmark$ | $\times$ | $\times$ | | textual | RelW |
| LogicBench (Parmar et al., 2024) | 2 | 2 | $\times$ | $\checkmark$ | $\checkmark$ | SN,DN | textual | RelW |
| SymTex | any | any | $\times$ | $\checkmark$ | $\checkmark$ | SN, DN, Disj,Cons | textual+ symbolic | RandW, RelW, RandS |

context-specific information becomes available (McCarthy, 1986; Brewka et al., 1997; Gigerenzer & Gaissmaier, 2011). Figure 1 shows a classical example of non-monotonic reasoning. In general, we consider "*the bird can fly*", but if given new information "*the bird is injured*", the original conclusion will be invalidated. Furthermore, if given another new information that "*the injury is slight*", "*the bird can fly*" will hold again. This example demonstrates how conclusions in reasoning can change with the addition of new information, highlighting the nature of non-monotonic reasoning.

Recently, the question of whether LLMs process logical reasoning capability, as well as the extent and nature of their reasoning ability, has received extensive attention, especially in non-monotonic reasoning. Xiu et al. (2022) created a pure non-monotonic reasoning dataset called LogicNMR, incorporating default rules. Parmar et al. (2024) introduced a benchmark, LogicBench, for evaluating the logical reasoning ability of LLMs, including non-monotonic reasoning. These works, including ours, focus on evaluating LLMs' ability to perform symbolic non-monotonic reasoning, which is the mainstream approach to non-monotonic reasoning (McCarthy, 1980; Reiter, 1980). In contrast, the studies in Rudinger et al. (2020) and Allaway et al. (2023) explore non-monotonic reasoning in natural language, driven by common-sense knowledge.

However, previous studies have overlooked several key factors: *(1)* They focus primarily on predicates with a single variable, even in multi-subject scenarios, limiting reasoning to one subject. *(2)* They have not thoroughly explored how the description of predicates affects LLMs' reasoning ability, which description of predicates is crucial for LLMs but not for traditional logic systems. *(3)* Most experiments are conducted in textual logic representation, without analyzing LLM performance in symbolic logic representation or comparing reasoning differences between symbolic and textual formats. Yet, textual logic representation inherently suffers from several disadvantages, such as ambiguities in natural language descriptions and challenges in conveying complex scenarios. These limitations are largely absent in symbolic logic representation, which offers a more precise and structured approach to representing information.

To fill these voids, we introduce a framework called **M**ulti-step **G**eneration for **Sym**bolic and **Tex**tual NMR Samples (MG-SymTex) designed to generate a non-monotonic dataset, referred to as SymTex. The MG-SymTex framework follows a three-step process: generation, modification, and textualization. The dataset, SymTex, is divided into 6 sub-datasets based on description types and predicate types. The differences between SymTex and related datasets are shown in Table 1. To ensure a fair evaluation between the symbolic and textual datasets, we structured the SymTex such that symbolic and textual samples correspond one-to-one within each predicate type. There are two tasks defined in SymTex: *(1)* Tri-State Boolean Querying, where LLMs need to assign a label to a query, given facts and rules; *(2)* Answer Set Computation, where LLMs are required to predict all possible conclusions, given the facts and rules.

We utilize SymTex to explore three questions regarding the non-monotonic reasoning ability of LLMs:

(1) To what extent do LLMs perform effectively on non-monotonic reasoning?

(2) What is the performance gap of LLMs in non-monotonic reasoning between symbolic and textual representations?

(3) To what extent do predicate descriptions influence the non-monotonic ability of LLMs?

Through extensive experiments, we find *(1)* The non-monotonic reasoning capability of LLMs is limited, as they struggle with tasks requiring dynamic adjustments and revisions throughout the reasoning process; *(2) In SymTex, the LLMs' performance gaps between symbolic and textual logic representations are -13.0% in average F1 on Tri-State Boolean Querying, and -2.8% in average EM-F1 on Answer Set Computation;* *(3)* The extent to which predicate types influence reasoning capability varies across different LLMs. *(4) Symbolic and textual samples can potentially complement each other in LLMs' reasoning.* A comprehensive discussion of these findings and additional results is provided in Section 5.

We summarize the main contributions as follows:

- A novel framework, named MG-SymTex, is proposed to automatically generate diverse non-monotonic samples by different parameters. MG-SymTex supports diverse sample styles and logical operations.
- A benchmark, named SymTex, is generated using MG-SymTex, which encompasses two primary tasks regarding non-monotonic reasoning, namely Tri-State Boolean Querying and Answer Set Computation. Additionally, it includes a dedicated subset aimed at assessing the ability of LLMs to correctly retract prior conclusions when presented with new critical information.
- An extensive experimental evaluation is conducted to assess the performance of LLMs, illustrating their limitations in non-monotonic reasoning. All codes and datasets will be publicly available when the paper is accepted.

## 2 RELATED WORK

### 2.1 NON-MONOTONIC REASONING

Non-monotonic reasoning (NMR) refers to a type of reasoning in which conclusions drawn from a set of premises can be retracted when new information is introduced. McCarthy (1980) presented circumscription, limiting reasoning to known facts in non-monotonic scenarios. Reiter (1980) developed a logic for default reasoning, applying default rules to draw conclusions with incomplete information. Pearl (1988) explored non-monotonic reasoning with causal relations affecting belief updates. Lascarides & Asher (1993) interpreted discourse relations using defeasible rules from commonsense knowledge. Chen et al. (2010) demonstrated that, in the propositional case, non-monotonic reasoning can be represented as an equivalent answer set program.

### 2.2 MONOTONIC LOGICAL REASONING WITH LLMS

Recently, LLMs have shown a powerful ability in various monotonic logical reasoning tasks, such as Multi-Step Reasoning (Saha et al., 2023; Fu et al., 2023) and Commonsense Reasoning (Tian et al., 2023; Perak et al., 2024). However, LLMs also exhibit notable limitations in reasoning tasks. Wang et al. (2024b) showed that LLMs' understanding of fundamental reasoning rules lags significantly behind human capability. Similarly, Srivatsa & Kochmar (2024) explored the challenges LLMs face in solving math word problems, while Li et al. (2024) demonstrated that LLMs perform considerably worse than neural program induction systems in reasoning tasks. Wang et al. (2024a) illustrated that LLMs struggle with understanding TBox NI transitivity rules. Parmar et al. (2024) showed that LLMs do not perform well in logic reasoning, even though they are in single inference rule scenarios. In this work, we identify a significant limitation of LLMs in their difficulty with non-monotonic reasoning.

### 2.3 NON-MONOTONIC REASONING BENCHMARK FOR LLMS

To evaluate the non-monotonic reasoning of language models, Rudinger et al. (2020) built a non-monotonic inference dataset called $\delta$-NLI, which provides new information to influence the belief of conclusions; Brahman et al. (2021) constructed a dataset based on $\delta$-NLI, providing the rationale for

the impact of new information; Xiu et al. (2022) introduced a dataset named LogicNMR, consisting of textual non-monotonic reasoning samples; Leidinger et al. (2024) focused on whether LLMs can maintain stable belief in generics at the addition of new information using the dataset from Allaway et al. (2023). Our work focuses on comprehensively evaluating the pure non-monotonic reasoning ability of LLMs, including symbolic and textual logic representation.

### 2.4 LLMs AS LOGIC SOLVERS AND CODE EXECUTORS

Recently, code has been recognized as a powerful tool for LLMs (Yang et al., 2024b) to access and leverage external sources. Meanwhile, there has been growing interest in exploring the role of LLMs as logic solvers and code executors. For example, Feng et al. (2023) utilized LLMs as Prolog logic solvers to address parsing errors in logic programs. Similarly, Chen et al. (2024b) explored how to guide LLMs in simulating logic solvers to execute Propositional Logic or Satisfiability Modulo Theories (SMT) programs, using natural language, Z3Py (Moura & Bjørner, 2008), or SMT-LIB (Barrett et al., 2010). Additionally, Wang et al. (2024c) demonstrated that LLMs can serve as executors when generated Z3 programs fail during execution, and Lyu et al. (2024) explored the feasibility of using LLMs as Python code executors. Our work focuses on leveraging LLMs as ASP solvers.

### 3 NON-MONOTONIC REASONING

In this work, we employ the framework of Answer Set Programming (ASP) (Gelfond & Lifschitz, 1988; 1991) because it is one of the most popular mechanisms for non-monotonic reasoning. An ASP program is a set of rules of the following form:

$$\omega(\mathbf{x}) \leftarrow \alpha_1(\mathbf{x}_1), \dots, \alpha_m(\mathbf{x}_m), \text{not } \alpha_{m+1}(\mathbf{x}_{m+1}), \dots, \text{not } \alpha_n(\mathbf{x}_n) \tag{1}$$

where each $\alpha_i(\mathbf{x}_i)$ is a literal of the form $p(\mathbf{x}_i)$ (positive literal) or $\neg p(\mathbf{x}_i)$ (negative literal), and each $\mathbf{x}_i$ consists of variables and constants. In ASP, "not" and "$\neg$" are called the default negation and the classical negation (strong negation). An ASP program (rule) is ground if there are no variables. A fact is a ground rule with $n = 0$. We often write an ASP problem as a pair $(W, D)$ with $W$ a set of facts, and $D$ a set of rules.

For example, assuming the bird is named Tweety, the three ASP programs $P_i = (W_i, D), i = 0, 1, 2$, where

$$W_0 = \{\text{Bird}(\textit{Tweety})\}; W_1 = W_0 \cup \{\text{Injured}(\textit{Tweety})\}; W_2 = W_1 \cup \{\text{SlightlyInjured}(\textit{Tweety})\}$$
$$D = \{\text{CanFly(A)} \leftarrow \text{Bird}(A), \text{not Abnormal(A)};$$
$$\text{Abnormal(A)} \leftarrow \text{Injured}(A), \text{not SlightlyInjured}(A)\}$$

represents the scenario depicted in Figure 1. Initially since $W_0$ contains only "Bird(*Tweety*)", $P_0$ intuitively entails "CanFly(*Tweety*)". The new information "Injured(*Tweety*)" in $(W_1, D)$ triggers the second rule in $D$, entails "Abnormal(*Tweety*)", and invalidates the first rule in $D$. Finally the fact "SlightlyInjured(*Tweety*)" in $(W_2, D)$ invalidates "Abnormal(*Tweety*)", allowing "CanFly(*Tweety*)" to be inferred once again.

The semantics of ASP are characterized by the notion of answer sets, also known as stable models Gelfond & Lifschitz (1988). An answer set *S* of (*W*, *D*) satisfies the following properties (Baral, 2003):

- $W \subseteq S$: All facts in $W$ are included in the answer set $S$.
- For every rule $(\omega \leftarrow \alpha_1, \dots, \alpha_m, \text{not } \alpha_{m+1}, \dots, \text{not } \alpha_n) \in D$, if $\alpha_1, \dots, \alpha_m \in S$ and $\alpha_{m+1}, \dots, \alpha_n \notin S$, then $\omega \in S$. This ensures that the rules in $D$ are respected in $S$.

Following our running example, $P_0$ has a unique answer set $W_0 \cup \{\text{CanFly}(\textit{Tweety})\}$, $P_1$ has a unique answer set $W_1 \cup \text{Abnormal}(\textit{Tweety})$, and $P_2$ has a unique answer set $W_2 \cup \{\text{CanFly}(\textit{Tweety})\}$.

In general, an ASP program may have 0, 1, or multiple answer sets. However, for the purpose of this work, we only consider ASP programs that have a unique answer set. We left the more general case to future work.

The ASP paradigm has been implemented in several ASP solvers, e.g., DLV (Alviano et al., 2017) and Clingo (Gebser et al., 2012). In this work, the use the latest version of DLV, *dlv2*[1], to validate the correctness of the symbolic samples in SymTex. For each symbolic sample, we represent the corresponding ASP program using the syntax supported by *dlv2*. Detailed examples of these programs are provided in Appendix A.

## 4 MG-SYMTEX

We propose a dataset generation framework called Multi-step Generation for Symbolic and Textual NMR Samples (MG-SymTex), which consists of three key steps: generation, modification, and textualization. Figure 2 illustrates the complete framework of MG-SymTex, along with examples from each stage. The use of a synthetic dataset is to provide a controlled environment that allows for direct evaluation of models' non-monotonic reasoning abilities, while eliminating the influence of semantics on the results.

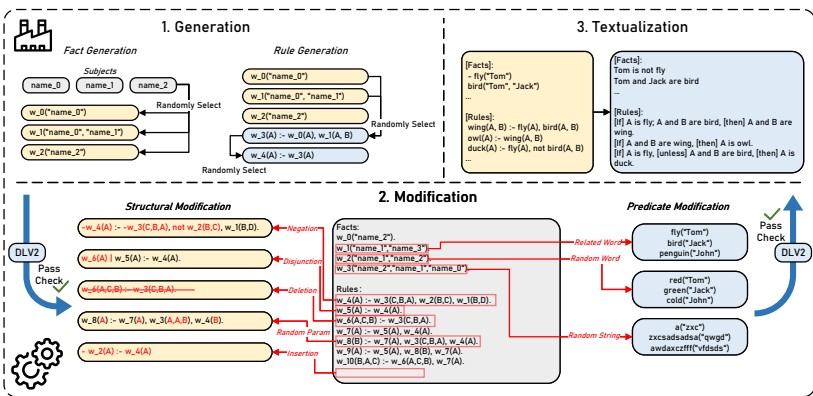

Figure 2: Overall framework and examples of MG-SymTex. The red parts are the modifications against the example in the previous stage.

### 4.1 GENERATION

In the generation phase, we aim to produce templates for use in the subsequent stages. These templates possess the following properties: *(1)* The description style of predicates and subjects is consistent, with subjects labeled as "name_i" for subject $i$ and predicates labeled as "w_i" for predicates $i$. *(2)* The templates are free from cyclic deductions. *(3)* No special operations, such as negation, disjunction, or constraints, are present in templates.

By first generating templates and then modifying them to create symbolic samples, the diversity of the dataset can be increased while reducing redundancy and similarity. The templates offer a consistent and structured foundation, ensuring well-defined logic and format.

The required parameters for the generation process are detailed in Table 2. To generate a template, start by creating $n_f$ facts using the parameters $max_{ap}$ and $max_{sub}$. For each fact, randomly select up to $max_{ap}$ terms (with replacement) from the set {name_i | $0 \leq i < max_{sub}$} to serve as arguments, and assign the predicate as $w_i$. Next, generate $n_r$ rules by selecting up to $max_{ar}$ terms from the set {w_i | $0 \leq i < n_f + j$} (including facts and the conclusions of previously created rules) to form the conditions of each rule. For each rule, randomly select up to $max_{ap}$ arguments, and assign the predicate w_$(n_r + j)$ as the rule's conclusion. To differentiate between rules and facts, convert arguments name_i into letters (e.g., name_0 → A, name_1 → B) for rules. The top left corner of Figure 2 shows examples for generating facts and rules.

---

[1] https://dlv.demacs.unical.it/

## 4.2 MODIFICATION

### 4.2.1 PROGRAM

After generating templates, we refine them through a modification process to produce symbolic samples in SymTex. Specifically, we employ 2 types of modification operations: structural modification and predicate modification.

As shown in the bottom of Figure 2, for structural modification, we introduce 6 different operations:

- Negation: Includes strong negation and default negation. The strong negation "-w" is true when the corresponding proposition "w" is explicitly false, while the default negation "not w" (also known as negation as failure) is true when there is no evidence to support the truth of "w". For example, "not w" can be true in two cases: either "-w" is explicitly asserted, or there is no available information regarding "w".
- Disjunction: Specifies a logical "or" operation, where at least one of the conclusions is true, denoted as "|".
- Deletion: Removes a fact or rule from the structure.
- Random Param: Adjusts the position of predicate parameters.
- Insertion: Adds a fact or rule to the structure. In this operation, a special logic construct called "constraint" may be introduced, represented as ":- A, B", which indicates that A and B cannot hold true simultaneously.

Moreover, for predicate modification, we introduce 3 various operations:

- Related Word: Uses a text encoder[2] to a build vector database by word along with its definition for WordNet (Miller, 1995). Then, randomly selects a word and identifies the top-k words with the highest cosine similarity to it as predicate descriptors. For subject descriptors, we utilize the Python library Faker[3] to generate random names.
- Random Word: The process is similar to that of "Related Word", but instead of choosing specific words, randomly selects k words to use as predicate descriptors.
- Random String: Randomly generates k strings as predicate and object descriptors.

Where k means the number of predicates in the given sample.

In practice, we first perform structural modification on templates, followed by predicate modification. During structural modification, each operation is assigned an independent probability of execution for each applicable fact and rule. Different operations can be executed simultaneously, depending on their respective probabilities. One template will be modified multiple times to build various samples. After modification, samples will be fed in *dlv2* for correctness verification.

### 4.2.2 ANSWER SET GENERATION

We conduct *dlv2* to execute the modified samples and corresponding templates to acquire answer sets $S_M$ and $S_T$, respectively. The labels of samples are built according to $S_M$ and $S_T$. For each atom $a \in S_M$:

$$L_a = \begin{cases} T, & a \in S_T \\ F, & \neg a \in S_T \\ M, & \text{otherwise} \end{cases} \tag{2}$$

Where $L_a$ is the label of $a$, and the $a \in S_M$ is negation-free.

## 4.3 TEXTUALIZATION

To generate textual samples that align one-to-one with symbolic samples, we use a template-based approach to create the corresponding textual datasets. A predefined linguistic template is utilized to map symbolic facts and rules to human-readable text. Conditions are placed after the "[if]"

---

[2]We utilize the *bge-m3* (Chen et al., 2024a) as the text encoder, which is available at `https://github.com/FlagOpen/FlagEmbedding`.

[3]`https://faker.readthedocs.io/en/master/`

Table 2: The parameters of the generation process.

| Parameter | Description |
|---|---|
| $n_f$ | The number of facts. |
| $n_r$ | The number of rules. |
| $max_{sub}$ | The maximum number of subjects in facts. |
| $max_{ar}$ | The maximum arity of rules. |
| $max_{ap}$ | The maximum arity of predicates. |

Table 3: Proportions (%) of various logical operations within the dataset samples.

| Predicate type | Strong Negation | Default Negation | Disjunction | Constraint |
|---|---|---|---|---|
| random string | 100 | 48.8 | 4.1 | 12.4 |
| random word | 100 | 50.6 | 4.1 | 12.4 |
| related word | 100 | 49.3 | 4.1 | 12.4 |

Table 4: The statistic for each subdatasets in SymTex.

| description type | predicate type | #samples | #queries | avg #labels | T:F:M |
|---|---|---|---|---|---|
| symbolic | random string | 28,780 | 120,775 | 4.2 | 0.8:0.5:1.7 |
| textual | random string | 28,780 | 120,775 | 4.2 | 0.8:0.5:1.7 |
| symbolic | random word | 28,180 | 118,214 | 4.2 | 0.8:0.5:1.7 |
| textual | random word | 28,180 | 118,214 | 4.2 | 0.8:0.5:1.7 |
| symbolic | related word | 28,434 | 119,073 | 4.2 | 0.8:0.5:1.7 |
| textual | related word | 28,434 | 119,073 | 4.2 | 0.8:0.5:1.7 |
| sum | - | 170,788 | 716,124 | - | - |

placeholder, and consequently follow the "[then]" placeholder. If default negation is present, it appears after the "[unless]" placeholder.

For each predicate, specific templates are applied: For single-argument predicates, the argument is placed before the "is" placeholder, and the predicate itself follows "is". For multi-argument predicates, the arguments are placed before the "are" placeholder. If negation is present, "not" is added after "is" or "are". For constraints, the conditions are framed between "It's not permissible for [" and "] to be true at the same time". The top right corner Figure 2 and Appendix B show examples for textualization.

### 4.4 SYMTEX

We employ the MG-SymTex framework to generate the SymTex dataset, as detailed in Figure 4. The statistic of SymTex is shown in Table 4, where SymTex comprises 6 sub-datasets, each with varying description and predicate configurations. In total, the dataset includes 170,788 samples and 716,124 queries. The proportions of various logical operations within SymTex are shown in Table 3. The examples of SymTex are shown in Appendix B. The differences between SymTex and other related datasets are shown in Table 1. Details for comparison between Symtex with others are shown in Appendix C, and the rule cover of SymTex is shown in Appendix D.

The SymTex is designed to support two primary tasks: Tri-State Boolean Querying and Answer Set Computation.

- **Tri-State Boolean Querying**: Given a program (facts, rules) and a query $q \in S_T$, the task is to determine the label of $q$, denoted as $L_q \in \{T, F, M\}$.
- **Answer Set Computation**: Given a program, the task is to generate the answer set $S_M$ while excluding the facts. The focus is on generating queries labeled as "T" or "F".

## 5 EXPERIMENTS

### 5.1 EVALUATION SETUP

#### 5.1.1 MODELS

To assess the zero-shot reasoning capability of LLMs using the SymTex dataset, we conducted experiments on 8 LLMs: *qwen2-7b* (Yang et al., 2024a), *mistral-7b* (Jiang et al., 2023), *llama3-8b* (AI, 2024), *gpt4o-mini* (OpenAI, 2024b), *gpt-4o* (OpenAI, 2024a), *claude-3-haiku* (Anthropic, 2024b), *claude-3-5-sonnet* (Anthropic, 2024a), and *o1-mini* (OpenAI, 2024c). For each task, we use the same prompt across all LLMs. The detailed prompts used in experiments are shown in Appendix E.

### 5.1.2 METRICS

For Tri-State Boolean Querying, we use the Marco-F1 (F1) score as the evaluation metric, while for Answer Set Computation, we adopt the exact match F1 (EM-F1).

For Tri-State Boolean Querying, each sample is assigned a single prediction label, and a prediction is considered correct only if it exactly matches the corresponding ground truth label. For Answer Set Computation, each sample is associated with a set of predicted answers, and a predicted answer is regarded as correct strictly when it matches an entry in the ground truth answer set without any discrepancies, including those in formatting, such as whitespace or case sensitivity.

All experiments are conducted with three independent runs, and we report the averaged results. Detailed findings and evaluations with additional metrics are provided in the Appendix G.1.

### 5.1.3 IMPLEMENTATION DETAILS

To ensure the stability of the output as much as possible, we set the temperature to 0 for all LLMs. To comprehensively evaluate various aspects of LLMs' reasoning capability, we derive different subsets from SymTex, with each subset generated through 3 independent runs, each comprising 1000 instances[4]. We construct 3 subsets as follows:

*(1)* SymTex$_{TBQ}$ aims to evaluate the overall reasoning ability of LLMs. It includes some samples that may not directly use default negation in reasoning, used to compare with Subset 2 which focuses directly on non-monotonic reasoning. *(2)* SymTex$_{TBQ}^{NM}$ aims to evaluate whether LLMs can change their prediction when facing information conflicting with default negation. *(3)* SymTex$_{ASC}$ aims to evaluate the LLMs' capability to solve ASP programs, which needs to generate all possible conclusions. Details for subset construction are shown in Appendix F.

## 5.2 MAIN RESULTS & ANALYSIS

We report the results of SymTex$_{TBQ}$, SymTex$_{TBQ}^{NM}$, and SymTex$_{ASC}$ in Table 5. Our main observations are summarized as follows:

**(1) To what extent do LLMs perform effectively on non-monotonic reasoning?** The non-monotonic reasoning capability of LLMs is limited.

In the Tri-State Boolean Querying task, although the powerful LLMs (*gpt-4o*, *claude-3.5-sonnet*, and *o1-mini*) achieve a high average F1 of nearly 80% in the textual settings of SymTex$_{TBQ}$ (Table 5), their performance in SymTex$_{TBQ}^{NM}$ dramatically decreases, especially in *claude-3.5-sonnet* whose average F1 drop from 80.8% to 54.9%. This indicates that LLMs struggle with tasks that require dynamic adjustments and revisions in reasoning processes.

In the Answer Set Computation task, from the results in Table 5, we observe that the smaller-scale LLMs perform extremely poor in the Answer Set Computation task (range of avg F1 between 0.2% and 1.9%), and similarly, the larger-scale LLMs also demonstrate relatively limited performance in this task (range of avg F1 between 10.8% and 40.6%). Although larger-scale LLMs show some improvement over smaller-scale LLMs, the enhancement is still inadequate given the Answer Set Computation task. This indicates that merely increasing the size of LLMs does not lead to substantial performance gains in Answer Set Computation, highlighting significant challenges that current LLMs face in handling this task.

**(2) What is the performance gap of LLMs in non-monotonic reasoning between symbolic and textual representations?** In the Tri-State Boolean Querying task, the LLMs' performance on the textual setting consistently outperforms that on the symbolic settings, where average F1 gains +13.0% and +9.7% improvement in SymTex$_{TBQ}$ (Table 5) and SymTex$_{TBQ}^{NM}$ (Table 5) respectively. In the Answer Set Computation task, the average EM-F1 in the textual setting outperforms that on symbolic, gaining +2.8% improvement (Table 5).

While LLMs exhibit stronger performance in textual settings, further analysis shows that symbolic and textual samples are complementary (Table 6). When used together for reasoning, LLMs have the

---

[4]For *o1-mini*, the temperature is set to 1, as this is the only supported configuration currently, and the sample number of subsets is set to 100.

potential to achieve more accurate answers, highlighting the importance of symbolic logic samples.

**(3) To what extent do predicate descriptions influence the non-monotonic ability of LLMs?**
Both in the Tri-State Boolean Querying and Answer Set Computation tasks, LLMs averagely perform better in random string settings, suggesting that the semantic information of predicates will impact the reasoning ability of LLMs.

In addition, Appendix G provides further experimental results, including the impact of Chain-of-Thought (Appendix G.3) and various temperature settings (Appendix G.4) for LLMs' capability of non-monotonic reasoning, along with a fine-grained analysis (Appendix G.2) and an in-depth error case study (Appendix G.5).

Table 5: Performance of LLMs on $\text{SymTex}_{\text{TBQ}}$, $\text{SymTex}_{\text{TBQ}}^{\text{NM}}$ and $\text{SymTex}_{\text{ASC}}$. *claude-3-H* and *claude-3.5-S* are the abbrevation of *claude-3-haik* and *claude-3.5-sonnet*. *Desc Types* means *description types*. *Sym* and *Tex* mean *symbolic* and *textual* respectively. *RandS*, *RandW* and *RelW* mean *random string*, *random word* and *related word*, respectively.

| Model | $\text{SymTex}_{\text{TBQ}}$ (F1) | | | | | $\text{SymTex}_{\text{TBQ}}^{\text{NM}}$ (F1) | | | | | $\text{SymTex}_{\text{ASC}}$ (EM-F1) | | | | |
| | Desc Types | | Predicate Types | | | Desc Types | | Predicate Types | | | Desc Types | | Predicate Types | | |
| | Sym | Tex | RandS | RandW | RelW | Sym | Tex | RandS | RandW | RelW | Sym | Tex | RandS | RandW | RelW |
| qwen2-7b | 33.2 | 38.6 | 41.3 | 31.2 | 35.2 | 29.6 | 39.0 | 37.2 | 32.1 | 33.7 | 1.9 | 1.9 | 1.7 | 1.5 | 2.5 |
| mistral-7b | 29.4 | 28.9 | 28.8 | 29.0 | 29.8 | 26.5 | 26.9 | 25.8 | 27.3 | 27.2 | 0.2 | 1.5 | 0.6 | 0.8 | 1.2 |
| llama3-8b | 31.3 | 47.1 | 39.3 | 38.4 | 39.9 | 26.6 | 37.0 | 31.4 | 33.3 | 30.8 | 1.3 | 0.4 | 0.3 | 0.9 | 1.4 |
| gpt-4o-mini | 36.5 | 57.6 | 49.0 | 46.0 | 46.2 | 24.5 | 41.9 | 34.2 | 33.6 | 31.9 | 11.8 | 12.7 | 13.3 | 10.7 | 12.8 |
| claude-3-haiku | 38.7 | 55.9 | 45.7 | 49.2 | 47.1 | 25.2 | 42.2 | 33.5 | 35.1 | 32.7 | 12.1 | 10.8 | 11.4 | 11.6 | 11.5 |
| gpt-4o | 51.6 | 73.7 | 63.5 | 62.0 | 62.5 | 39.5 | 63.5 | 53.7 | 50.7 | 50.2 | 20.9 | 26.2 | 24.8 | 22.8 | 23.1 |
| claude-3.5-sonnet | 73.0 | 80.8 | 75.5 | 78.7 | 76.5 | 54.9 | 53.9 | 52.6 | 57.4 | 53.4 | 30.0 | 25.6 | 31.1 | 24.4 | 28.0 |
| o1-mini | 58.0 | 81.5 | 74.5 | 67.8 | 67.0 | 51.3 | 51.2 | 58.3 | 47.8 | 47.7 | 19.4 | 40.6 | 31.6 | 30.0 | 28.5 |
| average | 44.0 | 58.0 | 52.2 | 50.3 | 50.5 | 34.8 | 44.5 | 40.8 | 39.6 | 38.4 | 12.2 | 15.0 | 14.3 | 12.8 | 13.6 |

## 5.3 VARIABLE IMPACT ANALYSIS ON RESULTS

To analyze the variables that significantly impact the results, we conduct experiments with varying numbers of query arity, related facts and rules, as well as noisy facts and rules. Specifically, the removal of related facts and rules alters the query label, whereas the removal of noisy facts and rules has no effect on the label.

From the results in Figure 3, the number of query arity, related facts, related rules, and noisy facts significantly impact the results. Increasing query arity, related facts, and related rules generally leads to a decrease in the F1 score. For noisy facts, their impact is small when fewer than 5, but becomes more significant with numbers greater than 5, resulting in a more noticeable decline in F1.

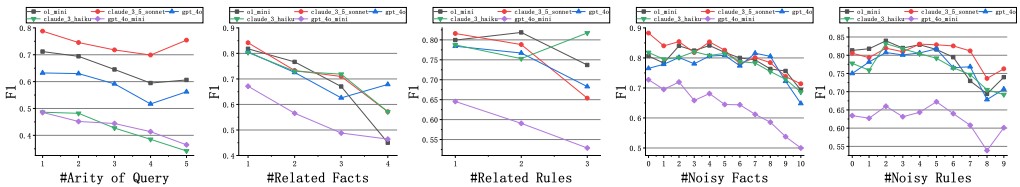

Figure 3: Results of different variables on $\text{SymTex}_{\text{TBQ}}$.

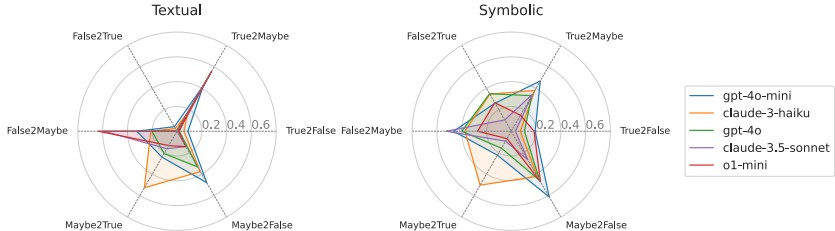

Figure 4: The fine-grained statistic of error samples on $\text{SymTex}_{\text{TBQ}}^{\text{NM}}$. The values are calculated by dividing the number of errors by the total number of true labels for each respective category. "X2Y" means the true label is X but the prediction is Y.

## 5.4 ERROR ANALYSIS

To investigate the specific sources of errors in non-monotonic reasoning exhibited by LLMs, we categorize the error types, as illustrated in Figure 4. We observe that in the textual setting, only a small number of errors result from the model misclassifying the label "T" as "F" or vice versa. The majority of errors are related to the label "M", occurring either when the true label is "M" or when the model incorrectly predicts "M". In the symbolic setting, although the performance of LLMs generally drops significantly, the majority of errors are still related to

Table 6: The percentage of correct/incorrect prediction combinations for different models. "S" and "T" mean in the symbolic and textual setting respectively. "C" and "I" mean correct and incorrect respectively.

| Models | S:C-T:C | S:C-T:I | S:I-T:C | S:I-T:I |
|---|---|---|---|---|
| claude-3.5-sonnet | 64.5 | 10.8 | 18.4 | 6.4 |
| gpt-4o | 44.4 | 10.2 | 30.1 | 15.3 |
| claude-3-haiku | 30.9 | 9.6 | 25.0 | 34.6 |
| gpt-4o-mini | 24.3 | 13.0 | 32.9 | 29.9 |

the label "M". This indicates that the inclusion of the label "M" significantly reduces the reasoning ability of LLMs, suggesting that LLMs struggle with handling ambiguous or complex cases represented by "M".

Additionally, as shown in Table 6, we compare how different models handle the same sample in symbolic and textual formats, examining the correct and incorrect prediction combinations in each format. The results show that the error rates in both formats are relatively low, suggesting that symbolic and textual formats are complementary. Using both formats together for reasoning could lead to more accurate answers, highlighting the importance of symbolic logic samples.

## 5.5 IMPACT OF NEW INFORMATION

To analyze the reaction when LLMs face new key information, we statistic the prediction shift on SymTex$_{TBQ}^{NM}$. From the results in Figure 5, we observe that introducing new information alters the predictions of the LLMs in 55.8% of the cases, and the accuracy of revised predictions is only 22.3%. Compared with different description types, the performance of LLMs in the symbolic setting is generally much better than that in the textual setting. The findings suggest that while LLMs are somewhat responsive to new information, they struggle to effectively incorporate it into correct predictions.

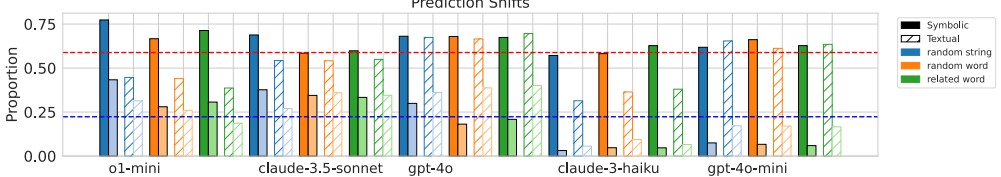

Figure 5: Prediction shifts of LLMs when exposed to new information on SymTex$_{TBQ}^{NM}$. The dark bars represent the proportion of predictions that change upon encountering new information, while the lighter bars represent the proportion of changes that lead to correct predictions. The red and blue dashed lines indicate the average values for these two cases, respectively.

## 6 CONCLUSION

In this work, we present MG-SymTex and introduce SymTex, a benchmark designed to evaluate LLMs' non-monotonic reasoning ability. SymTex includes diverse descriptions, predicates, and a rich set of logical operations. We define two key tasks: Tri-State Boolean Querying and Answer Set Computation, to rigorously assess LLM performance. Our experiments reveal significant limitations in current LLMs' ability to handle non-monotonic reasoning. Here are a few potential future research directions that could mitigate the aforementioned limitations: *(1)* Develop hybrid architectures that integrate symbolic logic representation with neural networks to leverage the strengths of both approaches; *(2)* Propose new innovative methods specifically tailored to enhance non-monotonic reasoning capability in LLMs; *(3)* Incorporate external modules, such as specialized reasoning agents, to augment and support the LLMs' inferential processes.

## 7 REPRODUCIBILITY

The codes and datasets for this work, including the construction of SymTex and its subsets, as well as the evaluation of LLMs, are provided in the Supplementary Material. To ensure reproducibility, we fix the random seed during each dataset construction step. Upon acceptance, all codes and datasets will be made publicly available.

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

## A   An Example of Dlv2

The symbolic sample of the scenario in Figure 1 is as follows:

```
Bird("Tweety").
Injured("Tweety").
SlightlyInjured("Tweety").
CanFly(A) :- Bird(A), not Abnormal(A).
Abnormal(A) :- Injured(A), not SlightlyInjured(A).
```

## B   Examples of SymTex

Figure 6, 7, and 8 illustrate SymTex under the random string, random word, and related word settings, respectively. Figure 9 presents examples of SymTex$_{\text{TBQ}}^{\text{NM}}$.

The examples for sample textualization are as follows:

- w_2(A) :- w_0(A), -w_1(A, B) → [if] A is w_0; A, and B are not w_1, [then] A is w_2.

- -w_2(A, B) :- w_0(A), not w_1(A, B) → [if] A is w_0; A, [unless] A, and B are not w_1, [then] A, and B are not w_2.

- :- w_0(A), -w_1(A, B) → It's not permissible for [ A is w_0; A, and B are not w_1 ] to be true at the same time.

---

**Random String**

Facts:
-rgKsgZXfFw("lxqsqwQscUv").
-rgKsgZXfFw("vFQjZouGaBkZo").
rgKsgZXfFw("LdDmrsnawOBER").
rgKsgZXfFw("qWSySc").

Rules:
-BjWlwwuq(A) :- rgKsgZXfFw(A).
sjJWHSK(A,A,A,A,A) :- ZzZhiFPBRyt(A), not -BjWlwwuq(A).
aTXNaZZOjXuxR(A) :- -JglmSOaqbBZquh(A), not -rbUuFVA(A).

Queries:
{"label": "M", "query": "BjWlwwuq(\"lxqsqwQscUv\")"},
{"label": "M", "query": "BjWlwwuq(\"vFQjZouGaBkZo\")"},
{"label": "F", "query": "BjWlwwuq(\"LdDmrsnawOBER\")"},
{"label": "F", "query": "BjWlwwuq(\"qWSySc\")"}

**Random String**

Facts:
lxqsqwQscUv is not rgKsgZXfFw.
vFQjZouGaBkZo is not rgKsgZXfFw.
LdDmrsnawOBER is rgKsgZXfFw.
qWSySc is rgKsgZXfFw.

Rules:
[If] A is rgKsgZXfFw, [then] A is not BjWlwwuq
[If] A is ZzZhiFPBRyt, [unless] A is not BjWlwwuq, [then] A, A, A, A and A are sjJWHSK
[If] A is not JglmSOaqbBZquh, [unless] A is not rbUuFVA, [then] A is aTXNaZZOjXuxR

Queries:
{"label": "M", "query": "lxqsqwQscUv is BjWlwwuq"}
{"label": "M", "query": "vFQjZouGaBkZo is BjWlwwuq"}
{"label": "F", "query": "LdDmrsnawOBER is BjWlwwuq"}
{"label": "F", "query": "qWSySc is BjWlwwuq"}

Figure 6: An example of SymTex in Random String setting.

**Random Word**

Facts:
see("Wendy_Perez","Joshua_Rodriguez","Daniel_Crane").
see("Joshua_Rodriguez","Victor_Olson","Amy_Golden").
-understand("Daniel_Crane").
cost("Wendy_Perez","Victor_Olson","Joshua_Rodriguez").
understand("Wendy_Perez").

Rules:
:- -part(B,A,A,D,D), not -part(B,A,A,D,D), good(E,D,A).
-good(D) :- understand(D), -see(E,A,B).
-top(A,A) :- understand(A).

Queries:
{"label": "M", "query": "good(\"Daniel_Crane\")"}
{"label": "M", "query": "good(\"Wendy_Perez\")"}
{"label": "M", "query": "top(\"Daniel_Crane\",\"Daniel_Crane\")"}
{"label": "F", "query": "top(\"Wendy_Perez\",\"Wendy_Perez\")"}

**Random Word**

Facts:
Wendy_Perez, Joshua_Rodriguez and Daniel_Crane are see.
Joshua_Rodriguez, Victor_Olson and Amy_Golden are see.
Daniel_Crane is not understand.
Wendy_Perez, Victor_Olson and Joshua_Rodriguez are cost.
Wendy_Perez is understand.

Rules:
It's not permissible for [E, D and A are good; B, A, A, D and D are not part; B, A, A, D and D are not part] to be true at the same time
[If] D is understand; E, A and B are not see, [then] D is not good
[If] A is understand, [then] A and A are not top

Queries:
{"label": "M", "query": "Daniel_Crane is good"}
{"label": "M", "query": "Wendy_Perez is good"}
{"label": "M", "query": "Daniel_Crane and Daniel_Crane are top"}
{"label": "F", "query": "Wendy_Perez and Wendy_Perez are top"}

Figure 7: An example of SymTex in Random Word setting.

**Related Word**

Facts:
-moderatism("Danielle_Berry","Mark_Benson").
moderatism("Todd_Bowen","Suzanne_Higgins").
-moderatism("Danielle_Berry","Danielle_Berry").

Rules:
-Town(B) :- -aesthetics(A), -technologist(A,B), not -activism(A,B).
-onyxis(A) :- -aesthetics(A).
hawkishness(B) :- -moderatism(A,B).
:- Town(B)

Queries:
{"label": "T", "query": "hawkishness(\"Danielle_Berry\")"}
{"label": "T", "query": "hawkishness(\"Mark_Benson\")"}
{"label": "M", "query": "hawkishness(\"Suzanne_Higgins\")"}

**Related Word**

Facts:
Danielle_Berry and Mark_Benson are not moderatism.
Todd_Bowen and Suzanne_Higgins are moderatism.
Danielle_Berry and Danielle_Berry are not moderatism.

Rules:
[If] A and B are not technologist; A is not aesthetics, [unless] A and B are not activism, [then] B is not Town
[If] A is not aesthetics, [then] A is not onyxis
[If] A and B are not moderatism, [then] B is hawkishness
It's not permissible for [B is Town] to be true at the same time

Queries:
{"label": "T", "query": "Danielle_Berry is hawkishness"}
{"label": "T", "query": "Mark_Benson is hawkishness"}
{"label": "M", "query": "Suzanne_Higgins" is hawkishness"}

Figure 8: An example of SymTex in Related Word setting.

Facts:
policy("James_Boyer").
policy("Mary_Harris").

Rules:
-catch(A) :- policy(A)
-ask(A) :- policy(A), not situation(A)
data(A) :- policy(A), -feeling(B)
situation(A) :- not -policy(A), data(A), catch(A)
right(A) :- catch(A), -situation(A)
garden(A) :- situation(A)
them(A)|-catch(A) :- -policy(A), -garden(A)

Query:
ask("James_Boyer")

Label:
False

Facts:
policy("James_Boyer").
policy("Mary_Harris").
**situation("James_Boyer")**

Rules:                           *contradict*
-catch(A) :- policy(A)
**-ask(A) :- policy(A), not situation(A)**
data(A) :- policy(A), -feeling(B)
situation(A) :- not -policy(A), data(A), catch(A)
right(A) :- catch(A), -situation(A)
garden(A) :- situation(A)
them(A)|-catch(A) :- -policy(A), -garden(A)

Query:
ask("James_Boyer")

Label:
**Maybe** ←                    *Invalidate*

Figure 9: Examples of SymTex$_{\text{TBQ}}^{\text{NM}}$.

Table 7: Common constructs of ASP programs.

| Construct | Explanation | Example | SymTex |
|---|---|---|---|
| Atoms | Basic facts or entities in the domain. | bird(sparrow) | √ |
| Literals | An atom or its negation. | fly(sparrow) or - fly(sparrow) | √ |
| Rules | Implications that define relationships between atoms (head :- body). | fly(X) :- bird(X), - penguin(X). | √ |
| Facts | Ground rules with no body, representing axioms. | bird(sparrow). | √ |
| Constraints | Rules without heads, used to restrict valid solutions. | :- fly(X), penguin(X). | √ |
| Choice Rules | Rules defining optional inclusion of atoms in answer sets. | {fly(X)} :- bird(X). | |
| Cardinality Constraints | Bounds on the number of satisfied literals. | 1 { fly(X) : bird(X) } 2. | |
| Aggregates | Functions (sum, count, min, max) applied to collections of literals. | totalWeight(W) :- W = #sum { weight(X) : selected(X) }. | |
| Negation as Failure | True if a literal cannot be proven true (negation by failure). | safe(X) :- not unsafe(X). | √ |
| Strong Negation | Classical negation, explicitly denoted by -. | -fly(X) :- penguin(X). | √ |
| Disjunctive Rules | Rules with multiple possible outcomes (disjunction in the head). | fly(X) \| swim(X) :- bird(X). | √ |
| Optimization Statements | Used to minimize or maximize an objective function. | #minimize { cost(X): selected(X) }. | |

## C  COMPARISON SYMTEX WITH OTHERS

The proposed dataset differs from existing ones in several ways, as summarized in Table 1.

Existing work such as $\delta$-NLI (Rudinger et al., 2020) does not focus on non-monotonic logic reasoning but rather on textual reasoning with non-monotonic situations; ProofWriter (Tafjord et al., 2021), ruletaker (Clark et al., 2021), and generics-exemplars Allaway et al. (2023) do not contain non-monotonic reasoning scenarios. Generics-exemplars only provide generic rules and exceptions (e.g., "Birds can fly, but penguins can't fly"), without actual reasoning scenarios. LogicNMR (Xiu et al., 2022) and LogicBench (Parmar et al., 2024) involve non-monotonic reasoning, but their logical structures are relatively simple, supporting only Default Negation and Strong Negation + Default Negation, respectively.

They also have limited predicate arguments (maximum of 1) and related objects (maximum of 2), which are insufficient for evaluating a model's reasoning abilities in more complex scenarios. The dataset proposed in this paper introduces a data generation framework that can construct facts and rules with arbitrary predicate arguments, supporting four types of logical operations: Strong Negation, Default Negation, Disjunction, and Constraints.

Moreover, other datasets typically use only one type of predicate description style (either Random Word or Related Word) during construction. Given that non-monotonic logic reasoning is independent of symbolic semantics, this dataset provides three predicate description styles—Random String, Random Word, and Related Word—to evaluate the model's sensitivity to different predicate descriptions in reasoning tasks.

## D  RULE COVER OF SYMTEX IN ASP

We have summarized and listed the ASP's constructs in Table 7. Our dataset covers most of the constructs of ASP programs and we support all the core features of ASP ("Negation as Failure" and "Disjunctive Rules"). Note that the constructs we do not support all belong to ASP extension extensions or syntax sugar.

# E  PROMPTS FOR TASKS

The prompts for classification and generation tasks are shown in Figure 10 and 11 respectively. Where the "{facts}" and "{rules}" are the corresponding component in the given sample; the "{response_format}" is different will the various description type of samples.

```
[Facts]:
{facts}

[Rules]:
{rules}

[Query]:
{query}

[task]:
Given a query and a set of facts and rules, determine the outcome by
evaluating the conditions specified. The possible outcomes are:

True: The query can be derived from the facts and rules.
False: The negation of the query can be derived from the facts and rules.
Maybe: Neither the query nor its negation can be derived from the facts
and rules.

The final conclusion should must in the following format:
<answer>True/False/Maybe</answer>
```

Figure 10: The prompt of classification tasks.

```
[Facts]:
{facts}

[Rules]:
{rules}

[task]:
Given a set of facts and rules, predicting all possible reasoning results for
True and False. Note that reasoning results can not be the facts.

True: The query can be derived from the facts and rules.
False: The negation of the query can be derived from the facts and rules.
{response_format}

[Response]:
```

```
Response format:
<true>
A("B");
A("B", "C");
</true>

<false>
-A("B");
-A("B", "C");
</false>

<summary>
A("B");
A("B", "C");
-A("B");
-A("B", "C");
</summary>
```
**Response format of symbolic sample**

```
Response format:
<true>
A is B;
A, B, and C are D;
</true>

<false>
A is not B;
A, B, and C are not D;
</false>

<summary>
A is B;
A, B, and C are D;
A is not B;
A, B, and C are not D;
</summary>
```
**Response format of textual sample**

Figure 11: The prompt for generation tasks.

# F  DETAILS FOR SUBSET CONSTRUCTION

*(1)* To assess the logical reasoning ability of LLMs, including both monotonic and non-monotonic, we introduce a subset called SymTex$_{TBQ}$. This subset is created by extracting facts, rules, and a query from SymTex samples. The motivation for evaluating the overall logical reasoning ability of LLMs, rather than focusing solely on non-monotonic reasoning, is to provide a comparative baseline. This contrast highlights the specific limitations of LLMs in non-monotonic reasoning, distinguishing them from their broader logical reasoning capability. *(2)* To specifically evaluate the non-monotonic reasoning capability of LLMs, we introduce a subset called SymTex$_{TBQ}^{NM}$. This subset consists of paired samples, where each pair includes: *(a)* Facts, rules, and a query labeled as "T" and "F"; *(b)* The same rules and query with a new fact to transfer the label from "T" or "F" to "M". An example for a paired sample is shown in Appendix B. *(3)* Additionally, we create a subset called SymTex$_{ASC}$, designed to assess the LLMs' ability to perform complex non-monotonic reasoning. This subset is generated by randomly selecting samples from SymTex. Table 8, 9 and 9 show the detailed results of Table 5.

# G    MORE EXPERIMENTAL RESULTS

## G.1    DETAIL RESULTS AND MORE METRICS

For Tri-State Boolean Querying, we use the following metrics for evaluation: F1 represents Macro-F1; wF1 is Weighted Macro-F1, with sample weight $1/|L_i|$; Acc refers to Accuracy; and wAcc is Weighted Accuracy, with sample weight $1/|L_i|$, where $|L_i|$ represents the number of labels associated with the sample $i$.

For Answer Set Computation, we use the following metrics for evaluation: E-F1 represents Macro-F1 for exact matches; E-C is the proportion of exact matches that contain at least one correct answer; F-F1 refers to Macro-F1 for fuzzy matches (with spaces removed and all characters in lowercase); F-C is the proportion of fuzzy matches that contain at least one correct answer; and #p represents the average number of predicted labels.

Table 8, 9 and 9 show the detailed results of Table 5.

Table 8: Performance of LLMs on SymTex$_{\text{TBQ}}$. The values in the bottom right corner of each cell indicate the standard deviation. The cells in blue and red mean the value below or above the median respectively. *claude-3-H* and *claude-3.5-S* are the abbreviation of *claude-3-haik* and *claude-3.5-sonnet*.

| model | random string | | | | random word | | | | related word | | | | avg | |
|---|---|---|---|---|---|---|---|---|---|---|---|---|---|---|
| | F1 | wF1 | ACC | wACC | F1 | wF1 | ACC | wACC | F1 | wF1 | ACC | wACC | wF1 | wACC |
| symbolic | | | | | | | | | | | | | | |
| qwen2-7b | $41.2_{0.4}$ | $40.8_{0.2}$ | $47.8_{0.7}$ | $43.9_{0.4}$ | $27.0_{1.3}$ | $27.2_{1.2}$ | $35.8_{1.2}$ | $35.1_{0.8}$ | $31.4_{0.6}$ | $31.4_{0.6}$ | $39.4_{1.8}$ | $37.7_{0.9}$ | 33.1 | 38.9 |
| mistral-7b | $29.4_{1.7}$ | $23.7_{2.1}$ | $51.2_{0.9}$ | $33.6_{1.6}$ | $29.4_{2.5}$ | $24.0_{2.9}$ | $51.2_{1.2}$ | $33.9_{2.0}$ | $29.5_{0.2}$ | $24.6_{0.3}$ | $51.0_{0.5}$ | $33.7_{0.5}$ | 24.1 | 33.7 |
| llama3-8b | $30.8_{1.5}$ | $31.8_{1.9}$ | $36.4_{0.9}$ | $36.7_{1.5}$ | $32.1_{0.4}$ | $33.3_{0.8}$ | $37.5_{0.3}$ | $37.9_{1.1}$ | $31.0_{0.1}$ | $32.7_{0.3}$ | $35.7_{0.7}$ | $37.1_{0.4}$ | 32.6 | 37.2 |
| gpt-4o-mini | $39.9_{0.3}$ | $44.3_{0.2}$ | $40.7_{0.3}$ | $46.1_{0.2}$ | $34.2_{2.6}$ | $37.4_{2.8}$ | $35.1_{2.4}$ | $39.4_{2.6}$ | $35.3_{2.0}$ | $39.5_{2.7}$ | $36.0_{1.8}$ | $41.8_{2.9}$ | 40.4 | 42.4 |
| claude-3-H | $38.3_{1.5}$ | $46.1_{1.2}$ | $40.2_{1.5}$ | $51.7_{0.9}$ | $39.4_{1.5}$ | $47.4_{1.4}$ | $41.2_{1.7}$ | $52.3_{1.1}$ | $38.4_{0.9}$ | $46.5_{0.7}$ | $40.0_{1.2}$ | $50.9_{1.0}$ | 46.7 | 51.6 |
| gpt-4o | $54.2_{0.7}$ | $57.6_{0.9}$ | $57.5_{0.9}$ | $58.9_{0.9}$ | $48.6_{2.2}$ | $52.2_{2.7}$ | $51.4_{1.8}$ | $53.7_{2.7}$ | $52.0_{1.9}$ | $56.1_{1.9}$ | $55.3_{2.2}$ | $57.3_{1.9}$ | 55.3 | 56.6 |
| claude-3.5-S | $70.8_{0.9}$ | $75.4_{1.1}$ | $72.3_{0.5}$ | $75.7_{1.2}$ | $75.0_{2.4}$ | $78.9_{2.4}$ | $77.1_{2.1}$ | $78.9_{2.4}$ | $73.1_{2.0}$ | $77.0_{2.6}$ | $76.3_{1.4}$ | $77.2_{2.6}$ | 77.1 | 77.3 |
| o1-mini | $66.7_{0.5}$ | $72.5_{0.9}$ | $67.9_{0.7}$ | $73.3_{0.8}$ | $51.8_{0.8}$ | $57.0_{1.3}$ | $53.8_{0.7}$ | $58.7_{1.3}$ | $55.6_{0.9}$ | $62.5_{0.7}$ | $58.0_{1.0}$ | $63.7_{0.3}$ | 64.0 | 65.2 |
| avg | 46.4 | 49.0 | 51.8 | 52.5 | 42.2 | 44.7 | 47.9 | 48.7 | 43.3 | 46.3 | 49.0 | 49.9 | - | - |
| textual | | | | | | | | | | | | | | |
| qwen2-7b | $41.4_{1.3}$ | $49.5_{1.0}$ | $42.4_{1.5}$ | $54.6_{1.0}$ | $35.4_{2.2}$ | $41.1_{2.5}$ | $37.6_{1.9}$ | $45.2_{2.3}$ | $39.0_{0.2}$ | $47.8_{0.4}$ | $39.7_{0.2}$ | $51.6_{0.6}$ | 46.1 | 50.5 |
| mistral-7b | $28.1_{1.0}$ | $22.9_{0.8}$ | $49.6_{1.7}$ | $32.9_{0.7}$ | $28.5_{0.7}$ | $23.7_{1.0}$ | $50.2_{0.6}$ | $33.6_{0.6}$ | $30.0_{0.6}$ | $24.6_{0.8}$ | $53.1_{1.1}$ | $34.6_{1.0}$ | 23.7 | 33.7 |
| llama3-8b | $47.8_{1.7}$ | $51.6_{1.6}$ | $48.8_{1.7}$ | $52.2_{1.4}$ | $44.7_{3.2}$ | $47.5_{2.9}$ | $46.9_{3.3}$ | $48.3_{2.9}$ | $48.7_{0.6}$ | $52.8_{0.8}$ | $51.1_{0.5}$ | $53.5_{0.6}$ | 50.6 | 51.3 |
| gpt-4o-mini | $58.0_{1.4}$ | $63.8_{1.7}$ | $56.8_{1.2}$ | $64.0_{1.7}$ | $57.8_{0.6}$ | $63.5_{1.3}$ | $57.8_{0.4}$ | $63.5_{1.7}$ | $57.1_{1.8}$ | $63.6_{1.7}$ | $56.8_{2.2}$ | $63.7_{1.6}$ | 63.6 | 63.7 |
| claude-3-H | $53.0_{0.6}$ | $61.7_{0.3}$ | $52.8_{0.4}$ | $64.7_{1.1}$ | $58.9_{0.5}$ | $66.7_{0.9}$ | $59.0_{0.4}$ | $68.1_{1.1}$ | $55.7_{3.0}$ | $64.6_{3.0}$ | $55.6_{3.0}$ | $66.1_{2.8}$ | 64.3 | 66.3 |
| gpt-4o | $72.8_{1.3}$ | $78.7_{1.1}$ | $73.0_{1.4}$ | $79.1_{1.1}$ | $75.3_{0.4}$ | $81.2_{0.3}$ | $76.2_{0.2}$ | $81.5_{0.4}$ | $72.9_{0.6}$ | $79.2_{0.9}$ | $74.2_{0.5}$ | $79.4_{0.9}$ | 79.7 | 80.0 |
| claude-3.5-S | $80.1_{0.1}$ | $81.6_{0.3}$ | $81.6_{0.4}$ | $81.3_{0.4}$ | $82.4_{1.3}$ | $82.4_{2.0}$ | $84.4_{1.3}$ | $82.0_{2.0}$ | $79.9_{2.5}$ | $80.3_{2.9}$ | $82.5_{2.0}$ | $79.8_{3.0}$ | 81.4 | 81.0 |
| o1-mini | $82.2_{0.6}$ | $82.8_{0.9}$ | $83.6_{0.6}$ | $82.5_{1.0}$ | $83.8_{1.2}$ | $84.1_{1.3}$ | $85.5_{0.7}$ | $83.8_{1.3}$ | $78.4_{1.3}$ | $80.1_{1.4}$ | $80.6_{1.1}$ | $79.7_{1.6}$ | 82.3 | 82.0 |
| avg | 57.9 | 61.6 | 61.1 | 63.9 | 58.4 | 61.3 | 62.2 | 63.3 | 57.7 | 61.6 | 61.7 | 63.6 | - | - |

Table 9: Performance of LLMs on SymTex$_{\text{TBQ}}^{\text{NM}}$.

| model | random string | | | | random word | | | | related word | | | | avg | |
|---|---|---|---|---|---|---|---|---|---|---|---|---|---|---|
| | F1 | wF1 | ACC | wACC | F1 | wF1 | ACC | wACC | F1 | wF1 | ACC | wACC | wF1 | wACC |
| symbolic | | | | | | | | | | | | | | |
| qwen2-7b | $33.7_{0.3}$ | $31.9_{0.5}$ | $39.8_{0.6}$ | $34.7_{0.6}$ | $25.1_{0.3}$ | $24.7_{0.3}$ | $32.9_{0.3}$ | $31.7_{0.7}$ | $30.1_{1.0}$ | $29.3_{1.3}$ | $36.2_{0.2}$ | $34.0_{0.9}$ | 28.6 | 33.5 |
| mistral-7b | $25.2_{1.2}$ | $20.4_{1.4}$ | $46.1_{0.7}$ | $32.1_{0.9}$ | $26.6_{0.9}$ | $22.2_{1.1}$ | $45.7_{0.3}$ | $32.6_{0.4}$ | $27.8_{1.1}$ | $24.4_{1.3}$ | $44.6_{1.0}$ | $33.3_{1.0}$ | 22.3 | 32.7 |
| llama3-8b | $25.4_{1.4}$ | $25.8_{1.3}$ | $32.4_{1.0}$ | $32.5_{0.8}$ | $28.4_{1.2}$ | $29.1_{1.1}$ | $33.4_{0.7}$ | $33.9_{0.3}$ | $26.1_{0.8}$ | $26.3_{0.6}$ | $32.4_{1.2}$ | $31.9_{0.8}$ | 27.1 | 32.8 |
| gpt-4o-mini | $25.2_{1.6}$ | $26.6_{1.9}$ | $25.3_{1.6}$ | $26.1_{2.0}$ | $25.0_{0.8}$ | $27.1_{0.9}$ | $24.2_{0.7}$ | $26.7_{0.9}$ | $23.2_{1.2}$ | $25.2_{1.1}$ | $22.4_{1.2}$ | $24.9_{0.9}$ | 26.3 | 25.9 |
| claude-3-H | $25.3_{0.4}$ | $28.6_{0.3}$ | $27.0_{1.0}$ | $30.5_{0.7}$ | $25.9_{1.3}$ | $29.3_{1.4}$ | $27.6_{1.6}$ | $30.9_{1.8}$ | $24.4_{0.6}$ | $27.6_{1.0}$ | $25.4_{0.4}$ | $28.9_{0.8}$ | 28.5 | 30.1 |
| gpt-4o | $44.4_{0.7}$ | $43.6_{1.3}$ | $48.2_{0.6}$ | $44.7_{0.8}$ | $36.9_{0.6}$ | $37.4_{0.5}$ | $37.9_{0.6}$ | $37.6_{0.5}$ | $37.2_{0.4}$ | $37.2_{0.6}$ | $38.8_{0.2}$ | $37.4_{0.5}$ | 39.4 | 39.9 |
| claude-3.5-S | $57.1_{0.9}$ | $57.6_{0.6}$ | $60.0_{1.0}$ | $57.2_{0.8}$ | $55.7_{1.7}$ | $55.0_{2.5}$ | $59.5_{1.0}$ | $54.9_{2.3}$ | $51.9_{1.8}$ | $51.2_{1.3}$ | $56.5_{0.9}$ | $51.4_{2.0}$ | 54.6 | 54.5 |
| o1-mini | $58.3_{5.1}$ | $59.8_{4.6}$ | $60.3_{4.2}$ | $59.7_{4.7}$ | $45.9_{8.9}$ | $47.9_{9.7}$ | $46.0_{10.1}$ | $47.7_{9.9}$ | $49.7_{3.5}$ | $50.7_{2.7}$ | $51.7_{3.5}$ | $51.3_{2.5}$ | 52.8 | 52.9 |
| avg | 36.8 | 36.8 | 42.4 | 39.7 | 33.7 | 34.1 | 38.3 | 37.0 | 33.8 | 34.0 | 38.5 | 36.6 | - | - |
| textual | | | | | | | | | | | | | | |
| qwen2-7b | $40.7_{1.1}$ | $45.9_{1.0}$ | $42.6_{0.7}$ | $49.8_{0.6}$ | $39.1_{0.7}$ | $42.8_{0.8}$ | $40.3_{0.5}$ | $45.3_{0.8}$ | $37.3_{1.7}$ | $42.0_{2.3}$ | $38.9_{1.6}$ | $45.0_{2.4}$ | 43.6 | 46.7 |
| mistral-7b | $26.3_{1.1}$ | $23.0_{1.2}$ | $42.0_{1.3}$ | $31.3_{1.3}$ | $27.9_{0.9}$ | $24.3_{1.0}$ | $45.3_{0.5}$ | $33.7_{0.5}$ | $26.6_{0.2}$ | $23.1_{0.2}$ | $44.1_{0.6}$ | $32.7_{0.2}$ | 23.5 | 32.6 |
| llama3-8b | $37.4_{0.5}$ | $38.9_{0.3}$ | $38.2_{0.9}$ | $38.7_{0.4}$ | $38.1_{0.6}$ | $39.4_{0.5}$ | $39.3_{0.7}$ | $40.0_{0.5}$ | $35.5_{0.7}$ | $37.0_{0.9}$ | $36.6_{0.3}$ | $36.9_{0.7}$ | 38.4 | 38.5 |
| gpt-4o-mini | $43.1_{1.8}$ | $48.1_{1.9}$ | $42.2_{1.7}$ | $48.0_{2.2}$ | $42.1_{1.4}$ | $46.6_{1.7}$ | $41.2_{1.5}$ | $46.8_{1.7}$ | $40.5_{1.1}$ | $45.4_{1.2}$ | $39.6_{1.0}$ | $45.3_{1.7}$ | 46.7 | 46.7 |
| claude-3-H | $41.6_{0.9}$ | $49.3_{0.8}$ | $43.4_{0.5}$ | $54.9_{0.5}$ | $44.2_{1.0}$ | $51.1_{1.4}$ | $44.7_{1.2}$ | $55.2_{2.0}$ | $40.9_{1.1}$ | $48.6_{1.0}$ | $41.8_{1.1}$ | $53.1_{1.4}$ | 49.7 | 54.4 |
| gpt-4o | $62.9_{1.4}$ | $68.6_{1.5}$ | $63.1_{1.1}$ | $70.0_{1.7}$ | $64.5_{1.0}$ | $69.2_{0.6}$ | $64.4_{0.9}$ | $70.1_{0.8}$ | $63.1_{1.8}$ | $67.4_{1.7}$ | $63.4_{1.9}$ | $67.6_{1.6}$ | 68.4 | 69.2 |
| claude-3.5-S | $48.0_{0.6}$ | $47.4_{0.7}$ | $51.8_{0.4}$ | $46.9_{0.5}$ | $59.0_{1.2}$ | $58.4_{1.7}$ | $61.3_{0.8}$ | $57.5_{1.5}$ | $54.8_{1.9}$ | $53.9_{1.5}$ | $58.5_{2.1}$ | $53.4_{1.5}$ | 53.2 | 52.6 |
| o1-mini | $58.2_{3.1}$ | $57.6_{3.4}$ | $61.0_{3.5}$ | $57.1_{3.4}$ | $49.6_{2.8}$ | $47.5_{3.4}$ | $56.0_{1.0}$ | $48.4_{2.0}$ | $45.7_{2.3}$ | $44.3_{3.6}$ | $52.0_{1.0}$ | $44.9_{1.9}$ | 49.8 | 50.1 |
| avg | 44.8 | 47.4 | 48.0 | 49.6 | 45.6 | 47.4 | 49.1 | 49.6 | 43.1 | 45.2 | 46.9 | 47.4 | - | - |

Table 10: Performance of LLMs on SymTex$_{ASC}$. Avg F1 and C are the average values of both exact and fuzzy modes.

| model | random string | | | | | random word | | | | | related word | | | | | avg | |
|---|---|---|---|---|---|---|---|---|---|---|---|---|---|---|---|---|---|
| | E-F1 | E-C | F-F1 | F-C | #p | E-F1 | E-C | F-F1 | F-C | #p | E-F1 | E-C | F-F1 | F-C | #p | F1 | C |
| **symbolic** | | | | | | | | | | | | | | | | | |
| qwen2-7b | $2.1_{0.4}$ | $5.5_{0.9}$ | $2.3_{0.5}$ | $6.0_{1.1}$ | $3.9_{0.2}$ | $1.2_{0.3}$ | $3.5_{0.4}$ | $1.4_{0.3}$ | $3.9_{0.6}$ | $3.8_{0.0}$ | $2.3_{0.1}$ | $6.8_{0.3}$ | $2.5_{0.1}$ | $7.4_{0.2}$ | $3.4_{0.1}$ | 2.0 | 5.5 |
| mistral-7b | $0.2_{0.1}$ | $0.4_{0.3}$ | $0.2_{0.2}$ | $0.5_{0.4}$ | $2.7_{0.1}$ | $0.1_{0.1}$ | $0.3_{0.2}$ | $0.1_{0.1}$ | $0.3_{0.2}$ | $3.8_{0.0}$ | $0.3_{0.2}$ | $0.7_{0.6}$ | $0.3_{0.2}$ | $0.8_{0.5}$ | $3.9_{0.1}$ | 0.2 | 0.5 |
| llama3-8b | $0.3_{0.1}$ | $0.8_{0.2}$ | $0.4_{0.0}$ | $0.8_{0.1}$ | $2.3_{0.1}$ | $1.2_{0.1}$ | $2.7_{0.3}$ | $1.2_{0.1}$ | $2.7_{0.3}$ | $2.6_{0.0}$ | $2.4_{0.2}$ | $5.8_{0.6}$ | $2.6_{0.2}$ | $6.1_{0.7}$ | $3.1_{0.1}$ | 1.4 | 3.2 |
| gpt-4o-mini | $12.9_{0.4}$ | $35.8_{1.5}$ | $15.2_{0.3}$ | $41.7_{1.7}$ | $5.9_{0.1}$ | $9.8_{0.4}$ | $31.5_{1.2}$ | $11.6_{0.4}$ | $37.0_{1.4}$ | $6.8_{0.1}$ | $12.7_{0.2}$ | $42.6_{1.6}$ | $15.3_{0.5}$ | $50.0_{1.1}$ | $7.8_{0.0}$ | 12.9 | 39.8 |
| claude-3-H | $11.4_{0.3}$ | $34.5_{0.6}$ | $14.0_{0.5}$ | $41.5_{0.2}$ | $6.5_{0.1}$ | $12.5_{0.5}$ | $41.8_{0.5}$ | $14.9_{0.5}$ | $48.6_{0.9}$ | $8.2_{0.1}$ | $12.5_{0.5}$ | $43.4_{1.0}$ | $14.8_{0.2}$ | $50.6_{0.2}$ | $9.1_{0.1}$ | 13.4 | 43.4 |
| gpt-4o | $22.7_{0.4}$ | $45.0_{0.4}$ | $29.6_{1.0}$ | $57.0_{1.0}$ | $4.3_{0.0}$ | $19.2_{0.4}$ | $43.1_{1.6}$ | $26.5_{0.5}$ | $56.7_{2.0}$ | $4.8_{0.1}$ | $20.7_{0.7}$ | $45.2_{0.9}$ | $27.7_{1.1}$ | $58.7_{2.0}$ | $5.1_{0.1}$ | 24.4 | 51.0 |
| claude-3.5-S | $33.8_{1.0}$ | $66.6_{1.4}$ | $40.9_{1.4}$ | $77.9_{1.3}$ | $5.3_{0.1}$ | $24.3_{3.4}$ | $52.0_{7.7}$ | $35.0_{4.2}$ | $70.3_{8.7}$ | $5.0_{0.6}$ | $31.8_{1.4}$ | $65.1_{1.8}$ | $40.0_{1.4}$ | $78.2_{2.3}$ | $5.5_{0.1}$ | 34.3 | 68.4 |
| o1-mini | $19.2_{3.1}$ | $29.3_{5.7}$ | $22.6_{0.7}$ | $33.0_{2.0}$ | $2.1_{0.7}$ | $21.1_{2.0}$ | $35.7_{3.2}$ | $24.7_{3.4}$ | $40.3_{3.8}$ | $2.9_{0.6}$ | $17.9_{3.4}$ | $32.0_{3.0}$ | $22.0_{2.7}$ | $37.7_{1.5}$ | $2.4_{0.1}$ | 21.3 | 34.7 |
| avg | 12.8 | 27.2 | 15.7 | 32.3 | - | 11.2 | 26.3 | 14.4 | 32.5 | - | 12.6 | 30.2 | 15.7 | 36.2 | - | - | - |
| **textual** | | | | | | | | | | | | | | | | | |
| qwen2-7b | $1.3_{0.2}$ | $4.8_{0.8}$ | $2.3_{0.4}$ | $7.6_{1.2}$ | $5.7_{0.2}$ | $1.8_{0.3}$ | $6.5_{1.1}$ | $3.6_{0.3}$ | $12.1_{1.1}$ | $5.4_{0.1}$ | $2.6_{0.3}$ | $9.7_{1.5}$ | $4.3_{0.3}$ | $15.2_{1.1}$ | $5.6_{0.1}$ | 2.7 | 9.3 |
| mistral-7b | $1.0_{0.0}$ | $4.2_{0.6}$ | $1.2_{0.0}$ | $4.7_{0.6}$ | $5.7_{0.2}$ | $1.5_{0.1}$ | $8.7_{1.0}$ | $2.0_{0.0}$ | $11.8_{1.5}$ | $8.5_{0.6}$ | $2.0_{0.1}$ | $12.4_{0.5}$ | $2.2_{0.1}$ | $13.7_{0.7}$ | $9.2_{0.2}$ | 1.7 | 9.3 |
| llama3-8b | $0.2_{0.1}$ | $0.6_{0.3}$ | $0.2_{0.1}$ | $0.6_{0.3}$ | $3.3_{0.0}$ | $0.6_{0.2}$ | $1.6_{0.5}$ | $0.6_{0.2}$ | $1.6_{0.5}$ | $3.1_{0.1}$ | $0.4_{0.1}$ | $1.3_{0.3}$ | $0.4_{0.1}$ | $1.3_{0.3}$ | $3.0_{0.2}$ | 0.4 | 1.2 |
| gpt-4o-mini | $13.6_{0.4}$ | $36.6_{1.3}$ | $13.6_{0.4}$ | $36.6_{1.3}$ | $5.0_{0.0}$ | $11.6_{0.5}$ | $32.1_{1.1}$ | $11.6_{0.5}$ | $32.1_{1.1}$ | $4.9_{0.0}$ | $12.9_{0.5}$ | $35.8_{1.6}$ | $12.9_{0.5}$ | $35.8_{1.6}$ | $5.3_{0.1}$ | 12.7 | 34.8 |
| claude-3-H | $11.3_{0.4}$ | $39.7_{0.9}$ | $11.3_{0.4}$ | $39.7_{0.9}$ | $8.9_{0.0}$ | $10.6_{0.3}$ | $39.6_{1.0}$ | $10.6_{0.3}$ | $39.6_{1.0}$ | $9.4_{0.1}$ | $10.4_{0.3}$ | $37.8_{1.4}$ | $10.5_{0.3}$ | $37.8_{1.4}$ | $9.0_{0.1}$ | 10.8 | 39.0 |
| gpt-4o | $26.9_{0.7}$ | $67.1_{1.9}$ | $27.0_{0.7}$ | $67.2_{1.9}$ | $6.4_{0.1}$ | $26.3_{0.7}$ | $64.2_{1.9}$ | $26.3_{0.7}$ | $64.2_{1.9}$ | $6.0_{0.1}$ | $25.4_{0.4}$ | $63.6_{1.0}$ | $25.4_{0.4}$ | $63.6_{1.0}$ | $6.4_{0.1}$ | 26.2 | 65.0 |
| claude-3.5-S | $28.4_{1.0}$ | $79.2_{1.9}$ | $28.4_{1.0}$ | $79.2_{1.9}$ | $8.9_{0.1}$ | $24.4_{0.1}$ | $76.2_{0.8}$ | $24.4_{0.1}$ | $76.2_{0.8}$ | $10.3_{0.1}$ | $24.1_{0.8}$ | $74.8_{0.7}$ | $24.1_{0.8}$ | $74.8_{0.7}$ | $10.0_{0.1}$ | 25.6 | 76.7 |
| o1-mini | $43.9_{2.6}$ | $61.7_{4.7}$ | $44.0_{2.6}$ | $62.0_{4.4}$ | $2.9_{0.2}$ | $38.8_{2.6}$ | $57.3_{4.2}$ | $38.8_{2.6}$ | $57.3_{4.2}$ | $3.5_{0.2}$ | $39.1_{1.6}$ | $61.0_{3.6}$ | $39.1_{1.6}$ | $61.0_{3.6}$ | $3.3_{0.3}$ | 40.6 | 60.1 |
| avg | 15.8 | 36.7 | 16.0 | 37.2 | - | 14.5 | 35.8 | 14.7 | 36.9 | - | 14.6 | 37.1 | 14.9 | 37.9 | - | - | - |

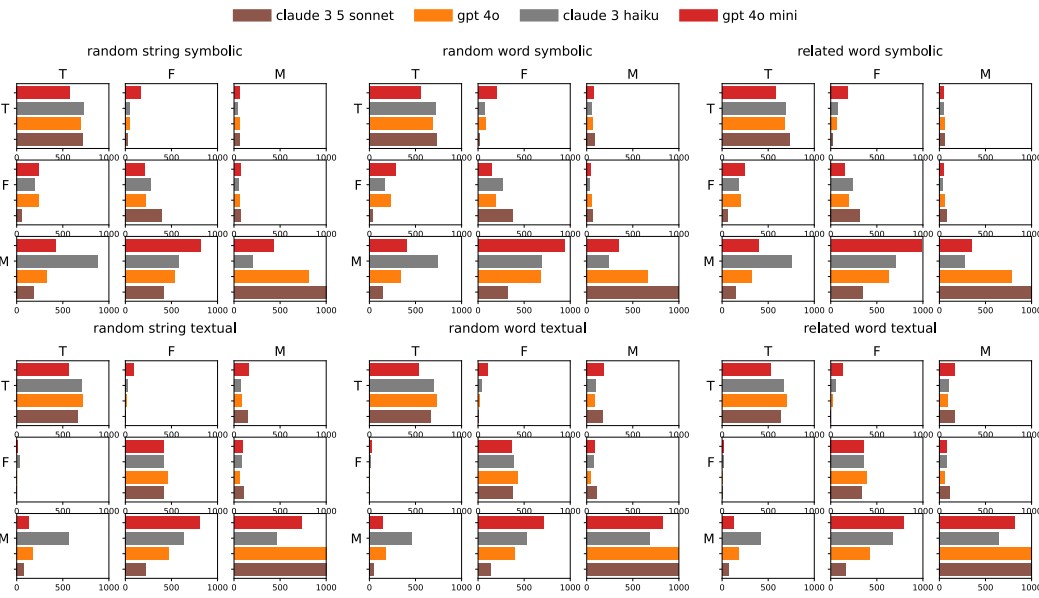

Figure 12: Confusion matrix for LLMs' predictions on SymTex$_{TBQ}$. The vertical axis represents true labels, and the horizontal axis represents predicted labels.

## G.2 FING-GRAINED ANALYSIS OF MAIN RESULTS

Figure 12 shows confusion matrix for LLMs' predictions on SymTex$_{TBQ}$. From the results, we observe that the superior performance of LLMs in the textual setting compared to the symbolic setting can be attributed to two factors: *(1)* In the textual setting, LLMs rarely misclassify samples with labels "T" or "F", whereas in the symbolic setting, samples with the label "F" exhibit a relatively high error rate; and *(2)* In the symbolic setting, the accuracy for samples with the label "M" is increased. Additionally, the performance gap between *gpt-4o* and *gpt-4o-mini* is primarily driven by discrepancies in the accuracy of the sample labeled "M". A similar pattern can be observed in the performance difference between *claude-3.5-sonnet* and *claude-3-haiku*. Figure 13 shows another perspective of the confusion matrix.

Figure 14 and 15 show confusion matrix for LLMs' predictions on SymTex$_{\text{TBQ}}^{\text{NM}}$. Compared with the results in Figure 12 and 13, the number of erroneous samples with true labels "T" and "F" increases significantly, as these are frequently misclassified as "M". This highlights the inherent challenges LLMs face when dealing with non-monotonic reasoning rules.

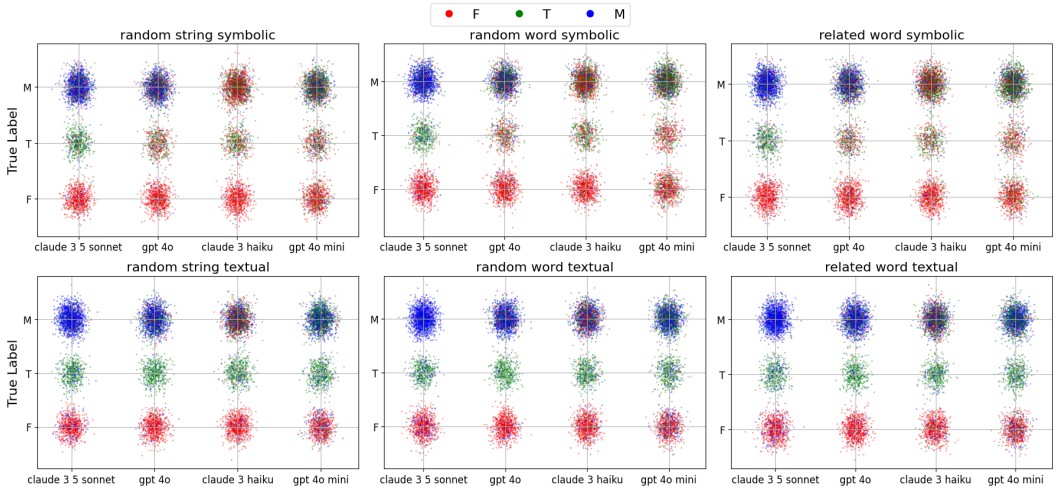

Figure 13: Scatter plot for LLMs' predictions on SymTex$_{\text{TBQ}}$. The vertical axis represents true labels, and the color of the scatters represents the predicted labels.

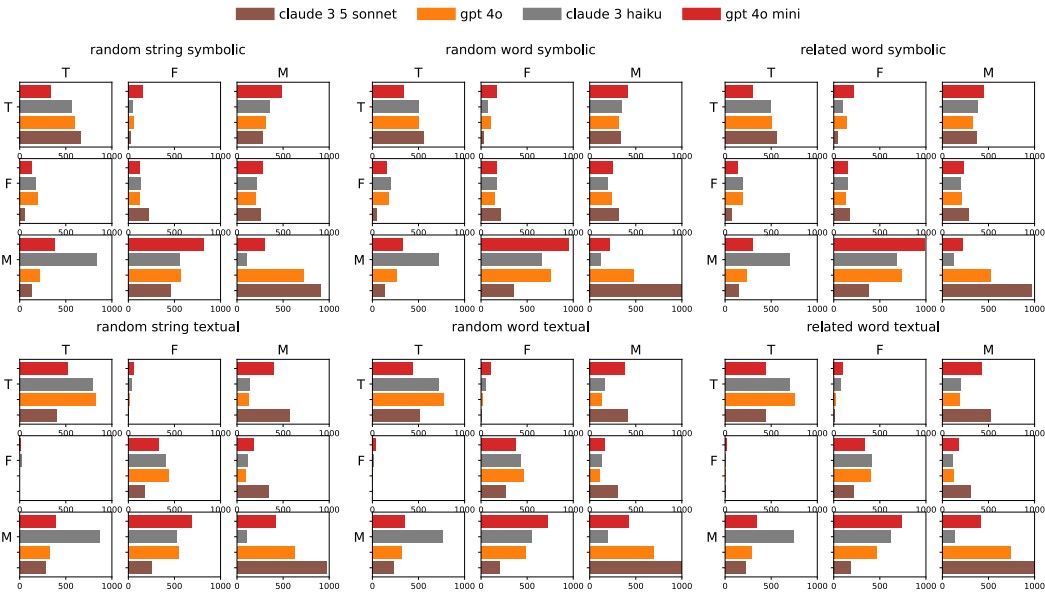

Figure 14: Confusion matrix for LLMs' predictions on SymTex$_{\text{TBQ}}^{\text{NM}}$.

## G.3 EFFECT OF CoT

Chain-of-Thought (CoT) (Wei et al., 2022) is a straightforward yet effective technique to improve LLMs' performance in reasoning and problem-solving. We perform experiments to assess the impact of CoT on SymTex. As the results shown in Table 11, in smaller-scale LLMs, the improvement

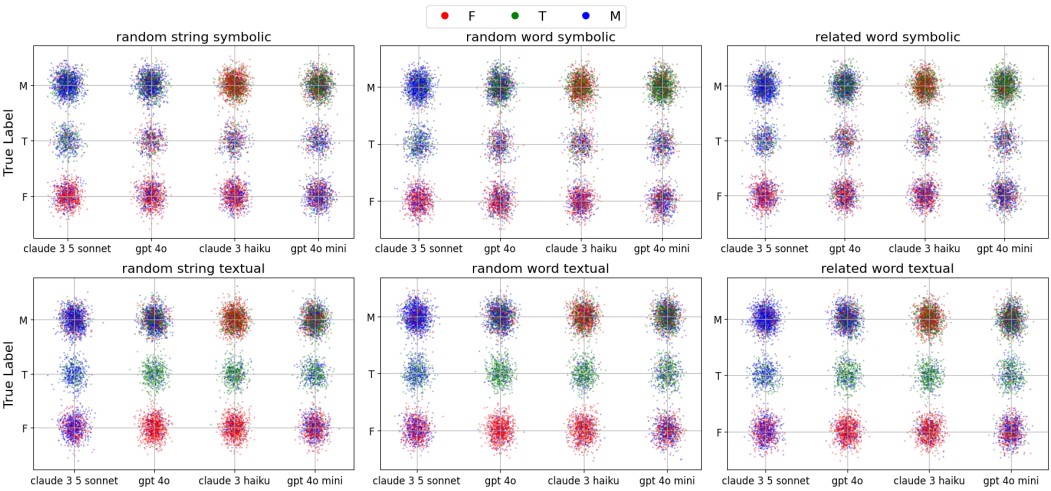

Figure 15: Scatter plot for LLMs' predictions on SymTex$_{TBQ}^{NM}$.

Table 11: The performance gap of LLMs on SymTex$_{TBQ}$ when using CoT versus without CoT.

| model | random string | | | | random word | | | | related word | | | | avg | |
|---|---|---|---|---|---|---|---|---|---|---|---|---|---|---|
| | F1 | wF1 | ACC | wACC | F1 | wF1 | ACC | wACC | F1 | wF1 | ACC | wACC | wF1 | wACC |
| **symbolic** | | | | | | | | | | | | | | |
| qwen2-7b | -2.0 | -2.7 | -1.3 | -2.8 | 12.4 | 8.7 | 15.3 | 5.1 | 10.7 | 7.3 | 14.8 | 4.9 | 4.4 | 2.4 |
| mistral-7b | 8.7 | 12.6 | -4.1 | 4.7 | 8.1 | 11.8 | -4.6 | 3.8 | 7.1 | 9.5 | -2.5 | 3.5 | 11.3 | 4.0 |
| llama3-8b | 0.3 | 0.6 | -0.3 | 0.5 | -0.6 | -0.7 | -0.4 | -0.7 | 0.3 | -0.2 | 1.1 | -0.2 | -0.1 | -0.1 |
| gpt-4o-mini | -0.9 | -1.3 | -1.0 | -1.2 | 0.6 | 0.9 | 0.8 | 1.2 | 0.5 | 0.1 | 0.5 | -0.3 | -0.1 | -0.1 |
| claude-3-haiku | -1.7 | -2.0 | -1.3 | -1.4 | -1.0 | -1.0 | -1.2 | -0.9 | -1.4 | -1.4 | -1.1 | -0.7 | -1.5 | -0.9 |
| gpt-4o | 1.9 | 1.8 | 2.3 | 1.5 | 2.0 | 2.1 | 2.1 | 1.8 | 1.0 | 0.5 | 1.4 | 0.5 | 1.5 | 1.3 |
| claude-3.5-sonnet | 0.7 | 0.6 | 0.7 | 0.6 | -0.3 | -0.1 | -0.4 | 0.0 | 0.4 | 0.8 | 0.2 | 0.8 | 0.4 | 0.5 |
| avg | 1.0 | 1.4 | -0.7 | 0.3 | 3.0 | 3.1 | 1.7 | 1.5 | 2.7 | 2.4 | 2.1 | 1.2 | - | - |
| **textual** | | | | | | | | | | | | | | |
| qwen2-7b | 12.0 | 3.1 | 16.8 | -1.4 | 15.1 | 6.6 | 21.8 | 4.0 | 15.5 | 5.6 | 22.2 | 2.3 | 5.1 | 1.6 |
| mistral-7b | 13.0 | 16.9 | -0.3 | 8.4 | 11.1 | 14.6 | -1.4 | 6.5 | 12.1 | 16.6 | -2.1 | 7.7 | 16.0 | 7.5 |
| llama3-8b | 3.7 | 1.2 | 5.7 | 0.4 | 3.6 | 2.1 | 5.2 | 1.7 | 1.6 | -1.5 | 4.3 | -1.9 | 0.6 | 0.1 |
| gpt-4o-mini | 1.1 | 1.2 | 1.3 | 1.2 | 0.4 | 0.4 | 0.5 | 0.5 | -0.7 | -0.2 | -1.2 | -0.1 | 0.5 | 0.5 |
| claude-3-haiku | -0.4 | 1.0 | -0.1 | 3.0 | -1.1 | -0.7 | -1.1 | -0.2 | 0.4 | 0.9 | 0.3 | 1.4 | 0.4 | 1.4 |
| gpt-4o | 3.0 | 2.2 | 3.3 | 2.0 | 2.5 | 1.7 | 2.7 | 1.6 | 1.7 | 1.2 | 1.8 | 1.1 | 1.7 | 1.6 |
| claude-3.5-sonnet | -0.3 | 0.3 | -0.5 | 0.2 | 0.3 | 1.6 | 0.2 | 1.7 | 0.1 | 1.6 | -0.4 | 1.7 | 1.2 | 1.2 |
| avg | 4.6 | 3.7 | 3.7 | 2.0 | 4.6 | 3.8 | 4.0 | 2.3 | 4.4 | 3.5 | 3.6 | 1.7 | - | - |

is significant on *qwen2-7b* (+4.8% on avg. wF1, +2.0% on avg. wAcc) and *mistral-7b* (+13.7% on avg. wF1, +5.8% on avg. wAcc), while the impact of CoT for *llama3-8b* (+0.2% on avg. wF1, +0.0% on avg. wAcc) is slight; in larger-scale LLMs, the impact of applying CoT or not is slight to the performance., where *gpt-4o-mini* gains +0.2% on both avg. wF1 and wAcc, and *claude-3-haiku* gains -0.5% and +0.2% on avg. wF1 and wAcc respectively.

Moreover, as shown in Table 12, the use of CoT in smaller-scale LLMs is less effective on this dataset compared to its performance on SymTex$_{TBQ}$. For certain models, such as *qwen2-7b*, CoT not only fails to improve performance but significantly degrades it. For larger-scale LLMs, the impact of applying CoT or not is still slight to the performance.

### G.4   IMPACT OF TEMPERATURE

To explore the impact of temperature on the non-monotonic reasoning capability of LLMs. We perform experiments across a range of temperature settings, specifically from 0.0 to 1.0, to comprehensively assess the variations in performance. During the experimental process, we construct smaller datasets from SymTex$_{TBQ}$ and SymTex$_{TBQ}^{NM}$, extracting 30 samples for each subset type. The smaller datasets are named Tiny-SymTex$_{TBQ}$ and Tiny-SymTex$_{TBQ}^{NM}$. Each experimental group runs 3 times, and we report the average results.

Table 12: The performance gap of LLMs on SymTex$_{TBQ}^{NM}$ when using CoT versus without CoT.

| model | random string | | | | random word | | | | related word | | | | avg | |
|---|---|---|---|---|---|---|---|---|---|---|---|---|---|---|
| | F1 | wF1 | ACC | wACC | F1 | wF1 | ACC | wACC | F1 | wF1 | ACC | wACC | wF1 | wACC |
| symbolic | | | | | | | | | | | | | | |
| qwen2-7b | -2.0 | -2.6 | -0.8 | -2.4 | 5.9 | 3.5 | 7.3 | 0.2 | -1.3 | -3.3 | 1.3 | -4.5 | -0.8 | -2.2 |
| mistral-7b | 4.6 | 6.6 | -4.5 | 0.2 | 4.3 | 6.1 | -4.9 | -0.2 | 3.2 | 3.7 | -2.5 | -0.5 | 5.5 | -0.1 |
| llama3-8b | 3.5 | 2.9 | 1.2 | -0.3 | 1.5 | 1.3 | 0.5 | -0.1 | 3.6 | 3.0 | 3.3 | 1.9 | 2.4 | 0.5 |
| gpt-4o-mini | -1.7 | -1.6 | -2.0 | -1.6 | -0.4 | -0.5 | -0.4 | -0.4 | 0.6 | 1.3 | 0.3 | 1.5 | -0.3 | -0.2 |
| claude-3-haiku | -1.7 | -2.1 | -1.8 | -2.4 | -0.6 | -0.5 | -0.8 | -0.7 | -0.5 | -0.6 | -0.5 | -0.6 | -1.1 | -1.3 |
| gpt-4o | 1.4 | 1.2 | 2.0 | 1.2 | 2.2 | 1.7 | 2.7 | 1.7 | 1.6 | 1.4 | 2.0 | 1.4 | 1.4 | 1.4 |
| claude-3.5-sonnet | -1.2 | -0.9 | -1.6 | -1.0 | 1.6 | 2.0 | 1.0 | 1.9 | 2.0 | 2.2 | 1.6 | 2.1 | 1.1 | 1.0 |
| avg | 0.4 | 0.5 | -1.1 | -0.9 | 2.1 | 1.9 | 0.8 | 0.3 | 1.3 | 1.1 | 0.8 | 0.2 | - | - |
| textual | | | | | | | | | | | | | | |
| qwen2-7b | -3.3 | -10.4 | 1.0 | -12.7 | -5.0 | -11.7 | 2.9 | -10.8 | -0.3 | -7.6 | 6.2 | -8.1 | -9.9 | -10.6 |
| mistral-7b | 10.4 | 12.9 | 0.9 | 6.2 | 10.6 | 13.4 | -1.2 | 5.3 | 10.9 | 14.3 | -1.8 | 5.5 | 13.6 | 5.7 |
| llama3-8b | 2.7 | 1.2 | 4.4 | 1.5 | -1.3 | -2.6 | 0.2 | -2.8 | 1.3 | -0.2 | 2.6 | 0.0 | -0.5 | -0.4 |
| gpt-4o-mini | 1.5 | 1.2 | 1.8 | 1.3 | 0.9 | 1.3 | 1.0 | 1.5 | 0.5 | 0.8 | 0.7 | 0.9 | 1.1 | 1.2 |
| claude-3-haiku | 0.3 | 0.6 | 1.1 | 2.1 | -1.7 | -1.7 | -1.7 | -1.9 | -0.6 | -0.5 | -0.5 | -0.2 | -0.5 | 0.0 |
| gpt-4o | 1.8 | 0.4 | 1.9 | -0.8 | 3.6 | 2.6 | 3.7 | 2.0 | 1.3 | 1.0 | 1.3 | 0.8 | 1.3 | 0.7 |
| claude-3.5-sonnet | 0.6 | 0.6 | 0.8 | 0.6 | 0.6 | 0.6 | 0.7 | 0.6 | 0.9 | 0.8 | 1.1 | 0.7 | 0.7 | 0.6 |
| avg | 2.0 | 0.9 | 1.7 | -0.3 | 1.1 | 0.3 | 0.8 | -0.9 | 2.0 | 1.2 | 1.4 | -0.1 | - | - |

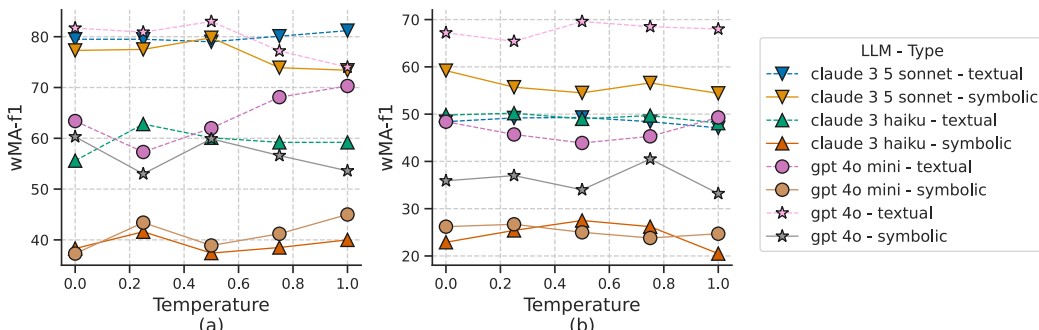

Figure 16: Results on various temperatures. (a) The results on Tiny-SymTex$_{TBQ}$; (b) The results on Tiny-SymTex$_{TBQ}^{NM}$.

From the results displayed in Figure 16, in Tiny-SymTex$_{TBQ}$, *gpt-4o-mini* and *claude-3-haiku* tend to exhibit improved performance with higher temperatures, while *gpt-4o* and *claude-3.5-sonnet* generally show a decline in performance under the same conditions. Furthermore, in Tiny-SymTex$_{TBQ}^{NM}$, LLMs appear to struggle to benefit from increased temperatures. This indicates that a higher temperature, which is claimed to bring greater creativity for LLMs, provides only limited benefits and may even impair the non-monotonic reasoning capability of LLMs.

### G.5 ERROR CASE ANALYSIS

To explore the reason behind the erroneous predictions of LLMs, we perform case studies on several examples, where both *claude-3.5-sonnet* and *gpt-4o* make incorrect label predictions. Figures 17 and 18 present detailed information about the samples, encompassing the facts, rules, queries, labels, and LLM responses.

From the error case 1 (Figure 17), we observe that *claude-3.5-sonnet* and *gpt-4o* both misunderstand the true meaning of the default negation. They view default negation as the same as strong negation in this case, which leads to incorrect interpretations and conclusions in contexts where the absence of evidence is not equivalent to the assertion of falsity.

From the error case 2 (Figure 18), we observe that *claude-3.5-sonnet* misinterprets the condition of default negation by treating it as a standard condition. Specifically, it incorrectly interprets the rule "[If] A and B are give, [unless] B and A are not receive, [then] B and A are news" as "[If] A and B are give; B and A are not receive, [then] B and A are news". This misunderstanding leads to an erroneous prediction, as the model fails to recognize the conditional dependency created by

Table 13: The confusion matrixes between actual and predicted labels for the classification task of S:I-T:C. The rows represent the actual labels (True Labels), while the columns represent the predicted labels.

| | gpt-4o | | | | claude-3.5-S | | |
|---|---|---|---|---|---|---|---|
| Label | F | T | M | Label | F | T | M |
| F | 0 | 603 | 136 | F | 0 | 116 | 114 |
| T | 172 | 0 | 137 | T | 61 | 0 | 119 |
| M | 1104 | 549 | 0 | M | 839 | 401 | 0 |

the default. Additionally, in this case, *gpt-4o* demonstrates a fundamental conceptual error in its approach to logical inference. Specifically, in classical logic, given a rule $\alpha \rightarrow \beta$, the presence of $\neg\alpha$ does not logically entail $\neg\beta$. However, *gpt-4o* incorrectly applies this reasoning, assigning the label "F" to the sample based on the assumption that blocking the rule negates the conclusion. This misinterpretation violates a core principle of logical reasoning and highlights a critical flaw in the LLM's handling of rule negation and implication.

Moreover, we statistic the results on gpt-4o and claude-3.5-Sonnet to analyze the gap between symbolic and textual settings. As shown in Table 13, The confusion matrixes reveal that both models share a common pattern, with errors concentrated in the case where the "True label is M, but the predication is F". The fact that errors mostly happen when the true label is "Maybe" but the model predicts "False" suggests that both models struggle with uncertainty or partial truths.

**Facts:**
Kathleen_Frazier is seem
Brenda_Bernard is seem
Darlene_Jackson is not seem

**Rules:**
[If] A is seem, [then] A is kitchen
[If] A is not seem, [unless] A is not piece, [then] A is not power
[If] A is not seem; A is not kitchen, [then] A is election
[If] A is seem; A is kitchen; A is not family; A is not election, [then] A is result
[If] A is family; A is not result, [then] A is not responsibility
[If] A is election; A is not seem; A is not family, [then] A is piece
[If] A is not seem; A is not result, [then] A is which
[If] A is responsibility, [then] A is tend
[If] A is tend; A is kitchen; A is not responsibility, [then] A is pattern

**Query:**
Darlene_Jackson is power

**Label:**
False

**Response:**
*Claude-3.5-sonnet:*
Let's evaluate this step by step: 1. From the facts, we know that Darlene_Jackson is not seem. 2. We can apply the rule: "[If] A is not seem, [unless] A is not piece, [then] A is not power" However, we don't know if Darlene_Jackson is piece or not piece, so we can't conclude anything about power yet. 3. We can apply the rule: "[If] A is not seem, [then] A is not kitchen, [then] A is election" Since Darlene_Jackson is not seem and not kitchen (as kitchen only applies to those who are seem), we can conclude that Darlene_Jackson is election. 4. Now we can apply the rule: "[If] A is election; A is not seem; A is not family, [then] A is piece" We know Darlene_Jackson is election and not seem, but we don't know about family. So we can't conclude if she is piece or not. 5. Since we can't determine if Darlene_Jackson is piece or not, we can't use the rule about power from step 2. 6. No other rules can help us determine if Darlene_Jackson is power or not power. Given this analysis, we cannot definitively say whether Darlene_Jackson is power or not power based on the given facts and rules. Therefore, the conclusion is: <answer>Maybe</answer>

*Gpt-4o:*
To determine if the query "Darlene_Jackson is power" is True, we need to evaluate the applicable rules and facts: 1. **Darlene_Jackson is not seem.** (Given fact) 2. **If Darlene_Jackson is not seem, unless Darlene_Jackson is not piece, then Darlene_Jackson is not power.** - To find if this rule applies, we need to assess if Darlene_Jackson is not piece. 3. **If Darlene_Jackson is not seem; Darlene_Jackson is not kitchen, then Darlene_Jackson is election.** - From the fact, Darlene_Jackson is not seem. We don't have a direct fact about Darlene_Jackson not being kitchen. - No conclusion yet about this rule because we can't confirm "not kitchen". 4. **If Darlene_Jackson is election; Darlene_Jackson is not seem; Darlene_Jackson is not family, then Darlene_Jackson is piece.** - This requires "Darlene_Jackson is election", which is not confirmed yet. 5. **If Darlene_Jackson is not seem; Darlene_Jackson is not result, then Darlene_Jackson is which.** - We can check if Darlene_Jackson is result or not. Since Darlene_Jackson is not seem, we only know it can't directly determine power without resolving whether it's a piece. We don't have enough information to confirm all conditions that determine "piece." Therefore, based on available facts and rules, we can't confirm "Darlene_Jackson is power" or its negation definitively. Thus, the outcome is: <answer>Maybe</answer>

Figure 17: Error case 1. The true label for the sample is "F", whereas both *claude-3.5-sonnet* and *gpt-4o* incorrectly predict it as "M".

**Facts:**
Ellen_Buck and Jessica_Morrison are give.
Ellen_Buck and Justin_Juarez are give.
Justin_Juarez and Ellen_Buck are not receive

**Rules:**
[If] A and B are give, [unless] B and A are not receive, [then] B and A are news
[If] B and A are news, [then] B and A are else
[If] A and A are receive; B and A are give, [then] B and A are field
[If] B is lawyer; B and A are field; B and A are not itself, [then] B is article

**Query:**
Justin_Juarez and Ellen_Buck are news

**Label:**
Maybe

**Response:**
*Claude-3.5-sonnet:*
Let's evaluate this step by step: 1. From the facts, we know: - Ellen_Buck and Jessica_Morrison are give - Ellen_Buck and Justin_Juarez are give - "Justin_Juarez" and "Ellen_Buck" are not receive 2. Looking at the first rule: [If] A and B are give, [unless] B and A are not receive, [then] B and A are news This rule applies to Justin_Juarez and Ellen_Buck because: - They are give (from the facts) - They are not receive (from the facts) 3. Therefore, we can conclude that Justin_Juarez and Ellen_Buck are news. 4. This directly matches our query: Justin_Juarez and Ellen_Buck are news Since we can derive this statement from the facts and rules, the query is true. <answer>True</answer>

*Gpt-4o:*
To determine the outcome, let's evaluate the given facts and rules: **Facts:** 1. Ellen_Buck and Jessica_Morrison are give. 2. Ellen_Buck and Justin_Juarez are give. 3. "Justin_Juarez" and "Ellen_Buck" are not receive. **Rules:** 1. If A and B are give, unless B and A are not receive, then B and A are news. 2. If B and A are news, then B and A are else. 3. If A and A are receive; B and A are give, then B and A are field. 4. If B is lawyer; B and A are field; B and A are not itself, then B is article. **Analyzing the Query:** - Query: Justin_Juarez and Ellen_Buck are news **Evaluation:** - From Rule 1: "If A and B are give, unless B and A are not receive, then B and A are news." - Ellen_Buck and Justin_Juarez are give (Fact 2). - Justin_Juarez and Ellen_Buck are not receive (Fact 3). Rule 1 applies because the condition "unless B and A are not receive" is met. Therefore, the conclusion "B and A are news" is blocked by the "unless" clause. Since Rule 1 is blocked and no condition in the facts or rules directly supports that Justin_Juarez and Ellen_Buck are news, the query cannot be confirmed. **Conclusion:** <answer>False</answer>

Figure 18: Error case 2. The true label for the sample is "M", whereas the predictions of *claude-3.5-sonnet* and *gpt-4o* are "T" and "F", respectively.

