# OpenReview forum: "SymTex: A New Benchmark for Non-monotonic Reasoning Capability of Large Language Models"
_ICLR.cc/2025/Conference — Submitted to ICLR 2025_

### Official Review · Reviewer_gF9G · 2024-10-27

**Soundness:** 2
**Presentation:** 3
**Contribution:** 2
**Rating:** 5
**Confidence:** 4

**Summary:**

This paper introduces SymTex, a benchmark for evaluating non-monotonic reasoning abilities of large language models. It also proposes a framework MG-SymTex, for generating non-monotonic reasoning samples in both symbolic and textual forms. The evaluation includes two tasks: tri-state boolean querying and answer set computation, and the authors analyze the performance of several state-of-the-art LLMs based on the proposed dataset.

**Strengths:**

(1) Non-monotonic reasoning is critical in logic as it allows models to revise conclusions when new information is introduced, thus the task is worth investigating but is currently underexplored.

(2) The process of creating the benchmark dataset is transparent and easy to follow.

(3) The paper provides detailed analyses for the LLMs' performance, including the impact of symbolic vs. textual representations, which highlights specific challenges faced by LLMs in NMR.

**Weaknesses:**

(1) The tasks designed in this benchmark seem highly artificial and do not reflect real-world non-monotonic reasoning challenges. For instance, the use of random strings or words as predicates (as in Figure 2) seems unrealistic for reasoning scenarios that LLMs encounter in natural language tasks. This limits the generalizability of the results. Real-world non-monotonic reasoning involves handling contextual and dynamic information, such as conversational changes, rather than simple logical queries over synthetic facts and rules. In other words, it would be necessary for the authors to justify the relevance of the proposed benchmark to real-world scenarios.

(2) The two proposed tasks—tri-state boolean querying and answer set computation—are not clearly defined in practical terms. The descriptions of both tasks lack details on **how the LLMs are supposed to process the input**, leading to confusion about what is being evaluated. For example, it is unclear how the "M" label is determined in Tri-State Boolean Querying, and why this label significantly complicates reasoning tasks. The evaluation metrics for the tasks (wF1, Acc, etc.) are briefly mentioned but not adequately explained. This weakens the reader's ability to assess the relevance of the tasks. More specific descriptions for the tasks should be provided, such as step-by-step examples of how an LLM should process a sample input for each task.

(3) The paper heavily focuses on the technical process of generating the SymTex dataset but does not sufficiently validate the quality or appropriateness of the generated samples. For example, the authors state that they use a tool called DLV2 to check the correctness of symbolic samples, but there is no discussion of how well the samples reflect actual non-monotonic reasoning problems.

Moreover, there is no attempt to benchmark the dataset against real-world datasets, making it hard to assess whether SymTex genuinely improves the evaluation of NMR in LLMs. It could make it better to compare SymTex against existing NMR datasets, or incorporate human experts to review the quality of the generated data samples.

(4) The novelty of approach is limited. Several recent works, such as $\delta$-NLI and LogicNMR, have introduced benchmarks for logical reasoning, including non-monotonic reasoning. The paper does not clearly differentiate itself from these works, except by offering symbolic samples, which still lacks clear justification. The limited novelty is also reflected in the shallow experimental results, where the findings mostly confirm known limitations of LLMs in handling complex logical tasks.

**Questions:**

The paper claims to analyze the gap between symbolic and textual reasoning but does not offer an in-depth analysis of why the performance differs significantly between the two. While it notes that there is a 13.5% performance drop in symbolic reasoning for tri-state boolean query, there is no deep dive into the causes. For example, why symbolic tasks pose such a challenge for LLMs is not explored sufficiently. Are the LLMs' errors due to the format of the data, the complexity of the logic, or limitations in their architecture?

---

> ### Author Response · Authors · 2024-11-20
> **Response - Part 1**
>
> We greatly appreciate the time and effort you have invested. In response to your concerns, we have provided clarifications here.
>
> > **W1**:The tasks designed in this benchmark seem highly artificial and do not reflect real-world non-monotonic reasoning challenges. For instance, the use of random strings or words as predicates (as in Figure 2) seems unrealistic for reasoning scenarios that LLMs encounter in natural language tasks. This limits the generalizability of the results. Real-world non-monotonic reasoning involves handling contextual and dynamic information, such as conversational changes, rather than simple logical queries over synthetic facts and rules. In other words, it would be necessary for the authors to justify the relevance of the proposed benchmark to real-world scenarios.
>
> **Response (W1):**
>
> (1): For the relation between the our focused task and natural language tasks.
>
> We would like to clarify that the focus of this work is different from the work given in [1] and [2], which consider non-monotonic reasoning in natural language driven by common-sense knowledge. **In contrast, our work considers evaluate the ability of LLMs on symbolic non-monotonic reasoning** **which is the main-stream of non-monotonic reasoning [3][4].**
>
> (2): For artifical data.
>
> **The purpose of this benchmark to use a synthetic dataset is due to the pure and controlled environment provided by this setting.** It can directly evaluate the non-monotonic reasoning ability of models and exclude the influence of semantics for evaluation results.
>
> (3): For the relation to real-world scenarios.
>
> **In symbolic non-monotonic reasoning, scenarios consist of facts and rules, and the process of dealing with given scenarios can be seen as dealing with dynamic information in the real world.** Moreover, we conducted experiments to assess whether LLMs can correctly modify the previous conclusion when given a new contradictory fact, as shown in the paper's Figure 4.
>
> **To better align with the reasoning habits of LLMs in real-world scenarios**, we also provide pairs of samples featuring symbolic and textual settings respectively.
>
> > **W2**: The two proposed tasks—tri-state boolean querying and answer set computation—are not clearly defined in practical terms. The descriptions of both tasks lack details on **how the LLMs are supposed to process the input**, leading to confusion about what is being evaluated. For example, it is unclear how the "M" label is determined in Tri-State Boolean Querying, and why this label significantly complicates reasoning tasks. The evaluation metrics for the tasks (wF1, Acc, etc.) are briefly mentioned but not adequately explained. This weakens the reader's ability to assess the relevance of the tasks. More specific descriptions for the tasks should be provided, such as step-by-step examples of how an LLM should process a sample input for each task.
>
> **Response (W2)**:
>
> (1): Definition of tasks.
>
> **The two tasks—tri-state boolean querying and answer set computation—are both about solving an ASP (Answer Set Programming) problem**. Tri-State Boolean Querying is a classification task that assigns a label to a query based on the facts and rules, while Answer Set Computation is a generation task that generates all possible conclusions from the given facts and rules. The label definition can be found in Section 4.2.2 ("ANSWER SET GENERATION").
>
> (2): Prompts for LLMs.
>
> The definition of prompts are shown in Appendix C ("PROMPTS FOR TASKS"), and we reported case study in Appendix D.4 ("ERROR CASE ANALYSIS").
>
> (3): Analysis of the label "Maybe".
>
> From the error case analysis in Appendix D.4 ("ERROR CASE ANALYSIS"), we discuss that LLMs always want to give a certain state to a conclusion. However, in non-monotonic reasoning, the conclusion may be in an uncertain state ("M"), which leads to error prediction from LLMs.
>
> (4): For metrics.
>
> In Section 5.1.2 ("METRICS"), we describe the definition and computation methods of the metrics. These metrics are commonly used in classification and question-answering tasks, so we provide only a brief description here. We plan to optimize them in subsequent versions.

---

> > ### Author Response · Authors · 2024-11-20
> > **Response - Part 2**
> >
> > > **W3**: The paper heavily focuses on the technical process of generating the SymTex dataset but does not sufficiently validate the quality or appropriateness of the generated samples. For example, the authors state that they use a tool called DLV2 to check the correctness of symbolic samples, but there is no discussion of how well the samples reflect actual non-monotonic reasoning problems.
> > >
> > > Moreover, there is no attempt to benchmark the dataset against real-world datasets, making it hard to assess whether SymTex genuinely improves the evaluation of NMR in LLMs. It could make it better to compare SymTex against existing NMR datasets, or incorporate human experts to review the quality of the generated data samples.
> >
> > **Response (W3)**:
> >
> > (1): For data quality.
> >
> > **This work focuses on scenarios that the non-monotonic reasoning logic programs can describe, especially by answer set programming (ASP).** In this case, we can evaluate the correctness of generated programs by executing them on ASP solvers.
> >
> >  (2): For data appropriateness and how the samples reflect real-world problems.
> >
> > To ensure diverse sample generation, we adjust various parameters, such as the number of facts and rules, as well as the maximum arity of rules and predicates. **This diversity in the generated logic programs allows us to simulate a wide range of real-world scenarios.**
> >
> > > **W4**: The novelty of approach is limited. Several recent works, such as δ-NLI and LogicNMR, have introduced benchmarks for logical reasoning, including non-monotonic reasoning. The paper does not clearly differentiate itself from these works, except by offering symbolic samples, which still lacks clear justification. The limited novelty is also reflected in the shallow experimental results, where the findings mostly confirm known limitations of LLMs in handling complex logical tasks.
> >
> > **Response (W4)**:
> >
> > (1): Difference between our proposed benchmark and previous related works.
> >
> > **The proposed dataset differs from existing ones in several ways, as summarized in Table Supplement.1**.
> >
> > Existing work such as δ-NLI [a] **does not focus on non-monotonic logic reasoning** but rather on textual reasoning with non-monotonic situations; ProofWriter [b], ruletaker [c], and generics-exemplars [e] **do not contain non-monotonic reasoning scenarios**. In fact, generics-exemplars [e] only provides generic rules and exceptions (e.g., "Birds can fly, but penguins can't fly"), **without actual reasoning scenarios**. LogicNMR [d] and LogicBench [f] involve non-monotonic reasoning, but their logical structures **are relatively simple**, supporting only Default Negation and Strong Negation + Default Negation, respectively.
> >
> > (2) New insights from our benchmark.
> >
> > Table Supplement.2 suggests that **if both formats are used together for reasoning, LLMs can potentially produce more accurate answers. This also underscores the necessity of symbolic logic samples.**
> >
> > > **Q1**: The paper claims to analyze the gap between symbolic and textual reasoning but does not offer an in-depth analysis of why the performance differs significantly between the two. While it notes that there is a 13.5% performance drop in symbolic reasoning for tri-state boolean query, there is no deep dive into the causes. For example, why symbolic tasks pose such a challenge for LLMs is not explored sufficiently. Are the LLMs' errors due to the format of the data, the complexity of the logic, or limitations in their architecture?
> >
> > (1): Analysis of the gap between symbolic and textual samples.
> >
> > Table Supplement.2 shows how the same sample is handled differently in symbolic and textual formats by different models, comparing the correct and incorrect combinations of predictions in both formats. The results suggest that the proportion of errors in both formats is not high, indicating that symbolic and textual samples are somewhat complementary. **If both formats are used together for reasoning, LLMs can potentially produce more accurate answers. This also underscores the necessity of symbolic logic samples.**
> >
> > (2): Analysis of the reason for the gap.
> >
> > We statistic the results on gpt-4o and claude-3.5-Sonnet, because they possess the largest and smallest performance gap between symbolic and textual samples. **As shown in Table Supplement.3, The confusion matrixes reveal that both models share a common pattern, with errors concentrated in the case where the "True label is M, but the predication is F"**. The fact that errors mostly happen when the true label is "Maybe" but the model predicts "False" suggests that both models struggle with uncertainty or partial truths.

---

> > > ### Author Response · Authors · 2024-11-20
> > > **Response - Part 3**
> > >
> > > ## Experiments
> > >
> > > Table Supplement.1: The difference between SymTex. "Commonsense- driven" means if commonsense knowledge is required for reasoning tasks in the dataset. In the **operations** column, abbreviations indicate supported logical operations: **SN (Strong Negation)**, **DN (Default Negation)**, **Disj (Disjunction)**, and **Cons (Constraint)**. Predicate style specifies the format of predicates within the dataset: **RamS (Random String)**, **RamW (Random Word)**, **RelW (Related Word)**. The **Multi-ary predicate** and **Multi-objects** columns indicate the maximum number of arguments and objects involved in reasoning tasks, respectively. If not specified, the value is marked as N/A.
> > >
> > > | **Dataset**               | **Multi-ary****predicate** | **Multi-objects** | **Commonsense- driven** | **Fact-rule-query** | **Non-monotonic** | **Operations**     | **Logic Style**  | **Predicate Style** |      |
> > > | ------------------------- | -------------------------- | ----------------- | ----------------------- | ------------------- | ----------------- | ------------------ | ---------------- | ------------------- | ---- |
> > > | **δ-NLI [a]**             |                            |                   | √                       | √                   | √                 |                    | textual          | RelW                |      |
> > > | **ProofWriter[b]**        | 2                          | 2                 | ×                       | √                   | ×                 | SN                 | textual          | RamW                |      |
> > > | **ruletaker[c]**          |                            |                   | ×                       | √                   | ×                 | SN                 | textual          | RamW                |      |
> > > | **LogicNMR[d]**           | 1                          | 1                 | ×                       | √                   | √                 | SN,DN              | textual          | RamW                |      |
> > > | **generics-exemplars[e]** |                            |                   | √                       | ×                   | ×                 |                    | textual          | RelW                |      |
> > > | **LogicBench[f]**         | 2                          | 2                 | ×                       | √                   | √                 | SN,DN              | textual          | RelW                |      |
> > > | **SymTex(ours)**          | any                        | any               | ×                       | √                   | √                 | SN, DN, Disj, Cons | textual+symbolic | RamW, RelW, RamS    |      |
> > >
> > >
> > > Table Supplement.2: This table shows the percentage of correct/incorrect prediction combinations for different models. "S" and "T" mean in the symbolic and textual setting respectively. "C" and "I" mean correct and incorrect respectively.
> > >
> > > |                       | **S:C_T:C** | **S:C_T:I** | **S:I_T:C** | **S:I_T:I** |
> > > | --------------------- | ----------- | ----------- | ----------- | ----------- |
> > > | **claude_3_5_sonnet** | 64.5        | 10.8        | 18.4        | 6.4         |
> > > | **gpt_4o**            | 44.4        | 10.2        | 30.1        | 15.3        |
> > > | **claude_3_haiku**    | 30.9        | 9.6         | 25.0        | 34.6        |
> > > | **gpt_4o_mini**       | 24.3        | 13.0        | 32.9        | 29.9        |
> > >
> > > Table Supplement.3: The table shows the comparison between actual and predicted labels for the classification task of S:I_T. The rows represent the actual labels (True Labels), while the columns represent the predicted labels.
> > >
> > > | **GPT-4o** | **False** | **True** | **Maybe** | **Claude-3.5-sonnet** | **False** | **True** | **Maybe** |
> > > | ---------- | --------- | -------- | --------- | --------------------- | --------- | -------- | --------- |
> > > | **False**  | 0         | 603      | 136       | **False**             | 0         | 116      | 114       |
> > > | **True**   | 172       | 0        | 137       | **True**              | 61        | 0        | 119       |
> > > | **Maybe**  | 1104      | 549      | 0         | **Maybe**             | 839       | 401      | 0         |

---

> > > > ### Author Response · Authors · 2024-11-20
> > > > **Response - Part 4**
> > > >
> > > > ## References
> > > >
> > > > [a] R. Rudinger, V. Shwartz, J. D. Hwang, and et al. Thinking like a skeptic: Defeasible inference in natural language. In Proc. of Findings of ACL’2020, pp. 4661–4675, 2020.
> > > >
> > > > [b] O. Tafjord, B. D. Mishra, & Clark, P. ProofWriter: Generating implications, proofs, and abductive statements over natural language. In Proc. of Findings of ACL/IJCNLP'2021, pp. 3621-3634, 2021.
> > > >
> > > > [c] P. Clark, O. Tafjord, & K. Richardson. Transformers as soft reasoners over language. In Proc. of IJCAI'2021, pp. 3882-3890, 2021.
> > > >
> > > > [d] Y. Xiu, Z. Xiao, and Y. Liu. Logicnmr: Probing the non-monotonic reasoning ability of pre-trained language models. In Proc. of Findings of EMNLP’2022, pp. 3616–3626, 2022.
> > > >
> > > > [e] E. Allaway, J. D. Hwang, C. Bhagavatula, and et al. Penguins don’t fly: Reasoning about generics through instantiations and exceptions. In Proc. of EACL’2023, pp. 2610–2627, 2023.
> > > >
> > > > [f] M. Parmar, N. Patel & N. Varshney. LogicBench: Towards systematic evaluation of logical reasoning ability of large language models. In Proc. of ACL'2024, pp. 13679-13707, 2024.
> > > >
> > > > [1] Y. Wang, H. Le, A. Gotmare, and et al. CodeT5+: Open Code Large Language Models for Code Understanding and Generation. In Proc. of EMNLP'2023, pp. 1069-1088, 2023.
> > > >
> > > > [2] D. Nam, A. Macvean, V. Hellendoorn, and et al. Using an llm to help with code understanding. In Proc. of the IEEE/ACM 46th International Conference on Software Engineering, pp. 1-13, 2024.
> > > >
> > > > [3] J. McCarthy. Circumscription—a form of non-monotonic reasoning. Artif. Intell., 13(1-2):27–39, 1980.
> > > >
> > > > [4] R. Reiter. A logic for default reasoning. Artif. Intell., 13(1-2):81–132, 1980.

---

> > > > > ### Comment · Reviewer_gF9G · 2024-11-23
> > > > >
> > > > > Thanks for the detailed responses, which have addressed most of my initial questions. I am inclined to raise my score once the proposed changes and additional content are reflected in the paper.
> > > > >
> > > > > Here are some further comments for consideration:
> > > > >
> > > > > (1) According to the authors’ response (W1), I understand the differences in formats between the proposed benchmark and existing works focusing on non-monotonic reasoning in natural language. However, that does not fully address my concern for the necessity of this benchmark. In other words, I am still suspicious about whether it is meaningful to evaluate LLMs with symbolic rules, and especially non-monotonic reasoning.
> > > > >
> > > > > I’ve noticed that other reviewers (rb9W, 9xu5) also have the same question, to which the authors responded that logic programs can be seen as a type of code, potentially linking this work to assessing LLM capabilities for code generation and understanding. However, to strengthen the argument, it would be beneficial if the authors could elaborate on real-world use cases, particularly (i) how **non-monotonic reasoning** is directly relevant to or impacts code generation tasks, and (ii) provide evidence or insights on whether there is a measurable correlation between logic reasoning abilities and code generation performance in LLMs. Such additional context would clarify the practical value of the work.
> > > > >
> > > > > (2) While the paper consistently uses the term "non-monotonic reasoning", the proposed benchmark focuses exclusively on **answer set programming**, which represents only a subset of non-monotonic reasoning. Therefore, I suggest revising the paper to emphasize this specific focus, such as narrowing the terminology to align with the actual scope of the benchmark. This adjustment would help avoid any ambiguity or overgeneralization in the paper's claims.
> > > > >
> > > > > Thanks again and I look forward to seeing these points addressed in the revised paper.

---

> > > > > > ### Author Response · Authors · 2024-11-27
> > > > > >
> > > > > > Thanks for your valuable feedback. In response to your concerns, we have provided clarifications here.
> > > > > >
> > > > > > > add.Q1:  (i) how non-monotonic reasoning is directly relevant to or impacts code generation tasks, and (ii) provide evidence or insights on whether there is a measurable correlation between logic reasoning abilities and code generation performance in LLMs.
> > > > > >
> > > > > > **Answer(add. Q1)**: Relation between non-monotonic reasoning to code generation and code understanding.
> > > > > >
> > > > > > **Answer Set Programming (ASP) is one of the most representative frameworks of non-monotonic reasoning, with ASP logic solvers like DLV [6] and Clingo [7] using code-style programs to represent facts and rules.** By generating correct code for non-monotonic reasoning problems, LLMs can leverage external logic solvers.  For example, the studies [8], [9], and [10] highlight that employing code as an intermediary between LLMs and logic solvers can significantly enhance the reasoning capabilities of LLMs.
> > > > > >
> > > > > > Moreover, understanding and reliably executing code for such problems by LLMs can enhance their internal logical reasoning capabilities. Recently, the works [11], [12], [13], and [14] investigated the potential of LLMs to function as logic solvers or code executors, leveraging the robustness of LLMs in handling minor grammatical or syntactic errors. Our work follows the same spirit of using LLMs to execute code in different formalisms.
> > > > > >
> > > > > > > add. Q2: While the paper consistently uses the term "non-monotonic reasoning", the proposed benchmark focuses exclusively on answer set programming, which represents only a subset of non-monotonic reasoning. Therefore, I suggest revising the paper to emphasize this specific focus, such as narrowing the terminology to align with the actual scope of the benchmark. This adjustment would help avoid any ambiguity or overgeneralization in the paper's claims.
> > > > > >
> > > > > >  **Answer (add. Q2)**: Relation between non-monotonic reasoning and answer set programming.
> > > > > >
> > > > > > ASP stands out as one of the most representative frameworks of non-monotonic reasoning. **In LogicNMR [15]**, although the authors claim that they use Default Logic as the NMR formalism, actually they have only used an (rather weak) ASP fragment of the Default logic. **In LogicBench [16]**, the studied NMR scenarios, including "Default Reasoning", "Reasoning about Unknown Expectations", and "Reasoning about Priorities", can all be effectively modeled by ASP.
> > > > > >
> > > > > > > Revised Paper
> > > > > >
> > > > > > **The revised paper has been submitted.**
> > > > > >
> > > > > > We sincerely appreciate the valuable feedback provided by the reviewers and have submitted the revised paper accordingly. We hope the revisions could address the reviewers' concerns, and we welcome any further questions or suggestions that may arise.
> > > > > >
> > > > > > ## Reference
> > > > > >
> > > > > > [6] M. Alviano, F. Calimeri, C. Dodaro, and et al. The ASP system dlv2. In Proc. of LPNMR'2017, pp. 215-221, 2017.
> > > > > >
> > > > > > [7] M. Gebser, R. Kaminski, B. Kaufmann, and et al. Theory solving made easy with Clingo 5. In Technical Communications of the 32nd International Conference on Logic Programming (ICLP 2016), 2016.
> > > > > >
> > > > > > [8] X. Yang, B. Chen, & Tam, Y. C. Arithmetic Reasoning with LLM: Prolog Generation & Permutation. In Proc. of NAACL'2024 (pp. 699-710), 2024.
> > > > > >
> > > > > > [9] T. Liu, W. Xu, W. Huang, and et al. Logic-of-thought: Injecting logic into contexts for full reasoning in large language models. arXiv preprint arXiv:2409.17539, 2024.
> > > > > >
> > > > > > [10] A. Kalyanpur, K. K. Saravanakumar, V. Barres, and et al. Llm-arc: Enhancing llms with an automated reasoning critic. arXiv preprint arXiv:2406.17663, 2024.
> > > > > >
> > > > > > [11] W. Wang, K. Liu, A. R. Chen, and et al. (2024). Python Symbolic Execution with LLM-powered Code Generation. arXiv preprint arXiv:2409.09271.
> > > > > >
> > > > > > [12] J. Feng, R. Xu, J. Hao, and et al (2023). Language models can be logical solvers. arXiv preprint arXiv:2311.06158.
> > > > > >
> > > > > > [13] M. Chen, G. Li, L. I. Wu, and et al. (2024). Can Language Models Pretend Solvers? Logic Code Simulation with LLMs. arXiv preprint arXiv:2403.16097.
> > > > > >
> > > > > > [14] C. Lyu, L. Yan, R. Xing, and et al. (2024). Large Language Models as Code Executors: An Exploratory Study. arXiv preprint arXiv:2410.06667.
> > > > > >
> > > > > > [15] Y. Xiu, Z. Xiao, and Y. Liu. LogicNMR: Probing the non-monotonic reasoning ability of pre-trained language models. In Proc. of Findings of EMNLP’2022, pp. 3616–3626, 2022.
> > > > > >
> > > > > > [16] M. Parmar, N. Patel & N. Varshney. LogicBench: Towards systematic evaluation of logical reasoning ability of large language models. In Proc. of ACL'2024, pp. 13679-13707, 2024.

---

> > > > > > > ### Comment · Reviewer_gF9G · 2024-11-27
> > > > > > >
> > > > > > > Thanks for the authors' further clarification.
> > > > > > >
> > > > > > > Following my concern on the necessity and usefulness of this benchmark, as I admit the added Section 2.4 which mentioned several works that apply LLMs as logic solvers or code executors, they do not constitute **direct evidence** to justify the necessity of an non-monotonic ASP benchmark for LLMs. Speaking about direct evidence, I would expect either (1) real-world use cases such as code generation that exactly match the form of ASP programs in the proposed benchmark; or (2) evidence showing that existing LLMs perform better on code generation if they perform better on SymTex.
> > > > > > >
> > > > > > > Given the current version of the paper and the above concerns, I decide to raise my score from 3 to 5.

---

> > > > > > > > ### Author Response · Authors · 2024-11-28
> > > > > > > >
> > > > > > > > Thanks for your valuable feedback. Here are some clarifications in response to your concerns.
> > > > > > > >
> > > > > > > > > **add.Q3**: (1) real-world use cases such as code generation that exactly match the form of ASP programs in the proposed benchmark;
> > > > > > > >
> > > > > > > > **Answer (add. Q3)**:
> > > > > > > >
> > > > > > > > In this benchmark, we aim to evaluate the capability of LLMs to execute ASP code accurately and reliably. **We consider that the reliable execution of code by LLMs is a prerequisite for effectively applying LLMs to solve real-world ASP problems.** Thus, our future work will focus on further exploring the application of LLMs in solving complex, real-world ASP cases, building on the foundation laid by this work.
> > > > > > > >
> > > > > > > > > **add.Q4**: (2) evidence showing that existing LLMs perform better on code generation if they perform better on SymTex.
> > > > > > > >
> > > > > > > > **Answer (add. Q4):**
> > > > > > > >
> > > > > > > > As clarified in Answer (add. Q1), SymTex is used to evaluate the capability of LLMs in executing ASP code, which differs from code generation tasks. **To assess whether LLMs that perform well on SymTex also excel in other tasks, we compute the Pearson correlation coefficient between performance on SymTex and that on other benchmarks.**
> > > > > > > >
> > > > > > > > For open-source LLMs, we use the average values from the Open LLM Leaderboard [17] as the reference. For closed-source LLMs, the global average and **coding average** values from LiveBench [18] are used as the reference.
> > > > > > > >
> > > > > > > > From the Pearson correlation coefficients presented in Table Supplement.4, Table Supplement.5, and Table Supplement.6, **it is evident that there is a notable relationship between the performance of SymTex and the performance on other benchmarks, including both overall ability and coding ability, for both open-source and closed-source LLMs.**
> > > > > > > >
> > > > > > > > ## Experiments
> > > > > > > >
> > > > > > > > Table Supplement.4: Pearson correlation coefficient of open-source LLMs between performance on SymTex and that on other benchmarks.
> > > > > > > > |                                     | Open LLM Leaderboard | Sym   | Tex   | SymTex avg. |
> > > > > > > > | ----------------------------------- | -------------------- | ----- | ----- | ----------- |
> > > > > > > > | Qwen/Qwen2-7B-Instruct              | 24.90                | 33.20 | 38.60 | 35.90       |
> > > > > > > > | mistralai/Mistral-7B-Instruct-v0.2  | 18.46                | 29.40 | 28.90 | 29.15       |
> > > > > > > > | meta-llama/Meta-Llama-3-8B-Instruct | 23.91                | 31.30 | 47.10 | 39.20       |
> > > > > > > > | Pearson correlation coefficient     | -                    | 0.93  | 0.81  | 0.89        |
> > > > > > > >
> > > > > > > > Table Supplement.5: Pearson correlation coefficient of close-source LLMs between performance on SymTex and that on other benchmarks.
> > > > > > > > |                                 | livebench | Sym   | Tex   | SymTex avg. |
> > > > > > > > | ------------------------------- | --------- | ----- | ----- | ----------- |
> > > > > > > > | gpt-4o-mini-2024-07-18          | 40.25     | 36.50 | 57.60 | 47.05       |
> > > > > > > > | claude-3-haiku-20240307         | 33.22     | 38.70 | 55.90 | 47.30       |
> > > > > > > > | gpt-4o-2024-08-06               | 53.77     | 51.60 | 73.80 | 62.70       |
> > > > > > > > | claude-3-5-sonnet-20240620      | 58.22     | 73.00 | 80.80 | 76.90       |
> > > > > > > > | o1-mini-2024-09-12              | 56.66     | 58.00 | 81.50 | 69.75       |
> > > > > > > > | Pearson correlation coefficient | -         | 0.88  | 0.98  | 0.94        |
> > > > > > > >
> > > > > > > > Table Supplement.6: Pearson correlation coefficient of closed-source LLMs between performance on SymTex and other **coding** benchmarks.
> > > > > > > > |                                 | livebench(coding) | Sym   | Tex   | SymTex avg. |
> > > > > > > > | ------------------------------- | ----------------- | ----- | ----- | ----------- |
> > > > > > > > | gpt-4o-mini-2024-07-18          | 43.51             | 36.50 | 57.60 | 47.05       |
> > > > > > > > | claude-3-haiku-20240307         | 24.46             | 38.70 | 55.90 | 47.30       |
> > > > > > > > | gpt-4o-2024-08-06               | 51.44             | 51.60 | 73.80 | 62.70       |
> > > > > > > > | claude-3-5-sonnet-20240620      | 60.85             | 73.00 | 80.80 | 76.90       |
> > > > > > > > | o1-mini-2024-09-12              | 48.05             | 58.00 | 81.50 | 69.75       |
> > > > > > > > | Pearson correlation coefficient | -                 | 0.80  | 0.81  | 0.82        |
> > > > > > > >
> > > > > > > > ## References
> > > > > > > >
> > > > > > > > [17] https://huggingface.co/spaces/open-llm-leaderboard/open_llm_leaderboard
> > > > > > > >
> > > > > > > > [18] C. White, S. Dooley, M. Roberts, and et al (2024). Livebench: A challenging, contamination-free LLM benchmark. arXiv preprint arXiv:2406.19314. https://livebench.ai/

---

> > > > > > > > > ### Comment · Reviewer_gF9G · 2024-11-28
> > > > > > > > >
> > > > > > > > > Thanks for the authors' response. As I understand the explanation that code execution by LLMs might be related to the ASP task, this still does not answer the question about "real-world use cases that exactly match the form of ASP programs". In other words, I cannot think about a case where I would need such a benchmark. Secondly, the positive correlation between SymTex and **other benchmarks** raises a new question that whether it is necessary to provide or use this benchmark. As a benchmark, it should be both (1) clearly related to some important, and practical task for LLMs, and (2) novel and cannot be replaced by existing benchmarks.
> > > > > > > > >
> > > > > > > > > Therefore, I would keep my current score.

---

> > > > > > > > > > ### Author Response · Authors · 2024-11-28
> > > > > > > > > > **Response - Part 1**
> > > > > > > > > >
> > > > > > > > > > Thanks for your feedback. We appreciate your willingness to engage in further discussion and raise thoughtful concerns. Here are some clarifications in response to your comments.
> > > > > > > > > >
> > > > > > > > > > > **add. Q5**: clearly related to some important, and practical task for LLMs
> > > > > > > > > >
> > > > > > > > > > **Answer (add. Q5):**
> > > > > > > > > >
> > > > > > > > > > This benchmark focuses on non-monotonic reasoning (NMR), which is crucial for logical reasoning—a key ability for LLMs.We use synthetic data due to the pure and controlled environment provided by this setting. **The datasets from ProofWriter [19], ruletaker [20], LogicNMR [21], and LogicBench [22] also rely on synthetic logic samples for similar reasons.**
> > > > > > > > > >
> > > > > > > > > > > **add. Q6**: novel and cannot be replaced by existing benchmarks.
> > > > > > > > > >
> > > > > > > > > > **Answer (add. Q6)**:
> > > > > > > > > >
> > > > > > > > > > From Table Supplement.1, we report the difference between SymTex and other related datasets.
> > > > > > > > > >
> > > > > > > > > > The novelty and irreplaceability of SynTex are as follows:
> > > > > > > > > >
> > > > > > > > > > (a) **SymTex supports any number of predicate arity and any number of objects used in reasoning**, whereas other datasets are limited to a maximum of 2 (or fewer).
> > > > > > > > > >
> > > > > > > > > > (b) **SymTex supports more NMR constructs (Strong Negation, Default Negation, Disjunction, Constraint)**, whereas other datasets only cover Strong Negation and Default Negation (or fewer).
> > > > > > > > > >
> > > > > > > > > > (c) **Symtex covers most of the constructs of ASP programs (Table Supplement.7) and we support all the core features of ASP ("Negation as Failure" and "Disjunctive Rules")**. Note that the constructs we do not support all belong to ASP extensions or syntax sugar. However, other datasets do not provide this level of coverage for ASP constructs.
> > > > > > > > > >
> > > > > > > > > > (d) **Our data generation framework will also be publicly available**, allowing other researchers to extend and analyze it. However, related works such as LogicNMR [21] and LogicBench [22] only provide the final datasets.
> > > > > > > > > >
> > > > > > > > > > > **add. Q7**: The positive correlation between SymTex and other benchmarks raises a new question of whether it is necessary to provide or use this benchmark.
> > > > > > > > > >
> > > > > > > > > > **Answer (add. Q7)** :
> > > > > > > > > >
> > > > > > > > > > Different datasets assess different abilities of LLMs, and models with stronger overall capabilities tend to perform better across multiple benchmarks. This is a general trend. As shown in Table Supplement.5 and Table Supplement.6, **there is also a significant positive correlation between the global average and coding average in LiveBench [18]. However, this does not imply that the benchmarks for coding are unnecessary.** Each benchmark provides various insights into specific aspects of model performance.
> > > > > > > > > >
> > > > > > > > > > ## Experiments
> > > > > > > > > >
> > > > > > > > > > Table Supplement.7 Common constructs of ASP programs.
> > > > > > > > > >
> > > > > > > > > > | **Construct**               | **Explanation**                                              | **Example**                                             | **SymTex** |
> > > > > > > > > > | --------------------------- | ------------------------------------------------------------ | ------------------------------------------------------- | ---------- |
> > > > > > > > > > | **Atoms**                   | Basic facts or entities in the domain.                       | bird(sparrow)                                           | √          |
> > > > > > > > > > | **Literals**                | An atom or its negation.                                     | fly(sparrow) or - fly(sparrow)                          | √          |
> > > > > > > > > > | **Rules**                   | Implications that define relationships between atoms (head :- body). | fly(X) :- bird(X), - penguin(X).                        | √          |
> > > > > > > > > > | **Facts**                   | Ground rules with no body, representing axioms.              | bird(sparrow).                                          | √          |
> > > > > > > > > > | **Constraints**             | Rules without heads, used to restrict valid solutions.       | :- fly(X), penguin(X).                                  | √          |
> > > > > > > > > > | **Choice Rules**            | Rules defining optional inclusion of atoms in answer sets.   | {fly(X)} :- bird(X).                                    |            |
> > > > > > > > > > | **Cardinality Constraints** | Bounds on the number of satisfied literals.                  | 1 { fly(X) : bird(X) } 2.                               |            |
> > > > > > > > > > | **Aggregates**              | Functions (sum, count, min, max) applied to collections of literals. | totalWeight(W) :- W = #sum { weight(X) : selected(X) }. |            |
> > > > > > > > > > | **Negation as Failure**     | True if a literal cannot be proven true (negation by failure). | safe(X) :- not unsafe(X).                               | √          |
> > > > > > > > > > | **Strong Negation**         | Classical negation, explicitly denoted by -.                 | -fly(X) :- penguin(X).                                  | √          |
> > > > > > > > > > | **Disjunctive Rules**       | Rules with multiple possible outcomes (disjunction in the head). | fly(X) \| swim(X) :- bird(X).                           | √          |
> > > > > > > > > > | **Optimization Statements** | Used to minimize or maximize an objective function.          | #minimize { cost(X): selected(X) }.                     |            |

---

> > > > > > > > > > > ### Author Response · Authors · 2024-11-28
> > > > > > > > > > > **Response - Part 2**
> > > > > > > > > > >
> > > > > > > > > > > ## references
> > > > > > > > > > >
> > > > > > > > > > > [18] C. White, S. Dooley, M. Roberts, and et al (2024). Livebench: A challenging, contamination-free llm benchmark. arXiv preprint arXiv:2406.19314. https://livebench.ai/
> > > > > > > > > > >
> > > > > > > > > > > [19] O. Tafjord, B. D. Mishra, & Clark, P. ProofWriter: Generating implications, proofs, and abductive statements over natural language. In Proc. of Findings of ACL/IJCNLP'2021, pp. 3621-3634, 2021.
> > > > > > > > > > >
> > > > > > > > > > > [20] P. Clark, O. Tafjord, & K. Richardson. Transformers as soft reasoners over language. In Proc. of IJCAI'2021, pp. 3882-3890, 2021.
> > > > > > > > > > >
> > > > > > > > > > > [21] Y. Xiu, Z. Xiao, and Y. Liu. LogicNMR: Probing the non-monotonic reasoning ability of pre-trained language models. In Proc. of Findings of EMNLP’2022, pp. 3616–3626, 2022.
> > > > > > > > > > >
> > > > > > > > > > > [22] M. Parmar, N. Patel & N. Varshney. LogicBench: Towards systematic evaluation of logical reasoning ability of large language models. In Proc. of ACL'2024, pp. 13679-13707, 2024.

---

> > > > > > > > > > > > ### Comment · Reviewer_gF9G · 2024-11-28
> > > > > > > > > > > >
> > > > > > > > > > > > Thanks for the responses. I'll keep my score.

---

### Official Review · Reviewer_9xu5 · 2024-10-30

**Soundness:** 3
**Presentation:** 3
**Contribution:** 3
**Rating:** 6
**Confidence:** 3

**Summary:**

The paper introduces the MG-SymTex framework, which can automatically generate a benchmark for non-monotonic reasoning problems with symbolic and textual representations. The framework consists of three stages: first, it generates a template; second, it modifies the template to create symbolic samples; and finally, it textualizes the modified samples. The paper evaluates several state-of-the-art LLMs and finds that they struggle with non-monotonic reasoning tasks. Additionally, it provides a detailed analysis of the extent to which LLMs perform non-monotonic reasoning, the performance gap between symbolic and textual representations, and the influence of predicate descriptions. Lastly, the paper includes an error analysis and examines how new information impacts LLMs' opinions.

**Strengths:**

(1)	The method introduces a novel framework capable of automatically generating a benchmark for non-monotonic reasoning tasks.

(2)	The paper presents extensive experiments to evaluate the ability of various state-of-the-art (SOTA) LLMs on this task.

(3)	The paper offers a comprehensive analysis of LLMs' performance on non-monotonic tasks, focusing on symbolic and textual representations, the impact of predicate descriptions and new information, and provides a detailed error analysis.

**Weaknesses:**

(1)	The motivation for testing LLMs' ability in the symbolic format requires further clarification. There are well-developed rule-based solvers for ASP problems that can solve ASP tasks effectively. Therefore, why focus on testing LLMs' ability on the ASP format instead of relying on rule-based solvers? In the analysis, you mention that improving LLMs' understanding of symbolic structures could enhance their ability to translate natural language into symbolic formats and leverage external solvers. However, solving ASP questions is not equivalent to translating natural language into ASP. Please correct me if my understanding is incorrect.

(2)	Further details are needed in Stage 1: Generation. How are the rules generated in this stage? There are various types of symbolic rules, so how do you determine which specific rule to use? Additionally, it is unclear why the modification is necessary. What purpose does modification serve when building the benchmark?

(3)	More details are needed for Stages 2 and 3. In Stage 2 (Predicate Modification), while you modify the predicate, it appears from Figure 2 that the argument (e.g., Tom, Jack) is also modified. However, in Stage 3 (Textualization), those specific arguments revert back to symbols (e.g., Tom becomes name_0). This process is somewhat confusing, and further elaboration is needed in the methodology to clarify the reasoning behind these steps.

(4)	More information is needed regarding the experimental setup. In Section 5.1.3, what is the main distinction between Subset 2 and Subset 3? Both seem to evaluate non-monotonic reasoning abilities, so why are there two separate subsets for this purpose? While I understand you may want to evaluate different aspects of the LLM, it would be helpful to explain the specific reason for using different subsets and how each one helps you achieve your goal, rather than simply describing how the subsets are constructed and letting the readers figuring it out themselves. Otherwise, readers might be unclear about the purpose of employing different subsets.

I will adjust my rating accordingly if the author addresses the concerns raised here.

**Questions:**

(1)	What is the main difference of your benchmark with the previous one? Is it with symbolic language? And what do you mean by pure non-monotonic and why the previous benchmarks are not pure in the introduction?

(2)	At stage 2 in your method, does structure modification refer to modifying the rules template generated at stage 1 and predicate modification refers to the facts from stage 1 as well?

(3)	Why is the argument in stage 2 modified to Tom and Jack but changed back to symbols at stage 3 according to Fig 2?

(4)	What is the purpose of using the three subsets in section 5?

---

> ### Author Response · Authors · 2024-11-20
> **Response - Part 1**
>
> We greatly appreciate the time and effort you have invested. In response to your concerns, we have provided clarifications here.
>
> > **W1**:The motivation for testing LLMs' ability in the symbolic format requires further clarification. There are well-developed rule-based solvers for ASP problems that can solve ASP tasks effectively. Therefore, why focus on testing LLMs' ability on the ASP format instead of relying on rule-based solvers? In the analysis, you mention that improving LLMs' understanding of symbolic structures could enhance their ability to translate natural language into symbolic formats and leverage external solvers. However, solving ASP questions is not equivalent to translating natural language into ASP. Please correct me if my understanding is incorrect.
>
> **Response (W1):**
>
> (1): The reason to evaluate the symbolic programs.
>
> **By evaluating the LLMs' ability in symbolic programs, we can gain a deeper understanding of the internal reasoning ability of LLMs**. Moreover, the logic programs, such as the ASP programs in our benchmark, is also a type of code, which is highly related to code generation or code understanding tasks on LLMs\[1\]\[2\], and non-monotonic reasoning has a well-established and mature symbolic foundation \[3\]\[4\].
>
> (2): Benefits when LLMs can execute ASP programs reliably.
>
> It's inconvenient to use ASP solvers in daily life, due to their complicated installation and strict grammar requirements. **If LLMs are reliable enough to infer on logic programs, they can be used as ASP solvers conveniently to deal with ASP problems, due to their tolerance for minor grammar and expression errors.**
>
> (3): Complement between symbolic and textual samples.
>
> Table Supplement.2 suggests that **if both formats are used together for reasoning, LLMs can potentially produce more accurate answers. This also underscores the necessity of symbolic logic samples.**
>
> > **W2**: Further details are needed in Stage 1: Generation. How are the rules generated in this stage? There are various types of symbolic rules, so how do you determine which specific rule to use? Additionally, it is unclear why the modification is necessary. What purpose does modification serve when building the benchmark?
>
> **Response (W2)**:
>
> (1): Rule generation.
>
> In Stage 1: Generation, symbolic rules are generated by randomly selecting predicates from a set of existing predicates to form the body of a new rule. A new predicate is then created to serve as the head of the rule.
>
> (2): There are various types of symbolic rules, so how do you determine which specific rule to use?
>
> We use the syntax of a state-of-the-art ASP (Answer Set Programming) solver, dlv2.
>
> (3): Purpose of modification.
>
> To avoid generating ASP programs with the same logical structure, we first create "templates" using only positive predicates. These templates provide a basic structure for the rules. Then, we modify them by conducting modification operations. This ensures the rules are more diverse and cover a wider range of scenarios.
>
> > **W3**: More details are needed for Stages 2 and 3. In Stage 2 (Predicate Modification), while you modify the predicate, it appears from Figure 2 that the argument (e.g., Tom, Jack) is also modified. However, in Stage 3 (Textualization), those specific arguments revert back to symbols (e.g., Tom becomes name_0). This process is somewhat confusing, and further elaboration is needed in the methodology to clarify the reasoning behind these steps.
> >
> > **Q3**: Why is the argument in stage 2 modified to Tom and Jack but changed back to symbols at stage 3 according to Fig 2?
>
> **Response (W3, Q3)**:
>
> Stage 3 will retain all the modifications from Stage 2. The description of the images was somewhat unclear, and we are preparing to revise it. Specific examples are shown in Appendix B ("SYMTEX Examples").

---

> > ### Author Response · Authors · 2024-11-20
> > **Response - Part 2**
> >
> > > **W4**: More information is needed regarding the experimental setup. In Section 5.1.3, what is the main distinction between Subset 2 and Subset 3? Both seem to evaluate non-monotonic reasoning abilities, so why are there two separate subsets for this purpose? While I understand you may want to evaluate different aspects of the LLM, it would be helpful to explain the specific reason for using different subsets and how each one helps you achieve your goal, rather than simply describing how the subsets are constructed and letting the readers figuring it out themselves. Otherwise, readers might be unclear about the purpose of employing different subsets.
> > >
> > > **Q4**: What is the purpose of using the three subsets in section 5?
> >
> > **Response (W4, Q4)**:
> >
> > - Subset 1 aims to **evaluate the overall reasoning ability of LLMs**. It includes some samples that may not directly use default negation in reasoning, used to compare with Subset 2 which focuses directly on non-monotonic reasoning.
> >
> > - Subset 2 aims to evaluate whether LLMs can **change their prediction** when facing information conflicting with default negation.
> >
> > - Subset 3 aims to evaluate the LLMs' **capability to solve ASP programs**, which needs to generate all possible conclusions.
> >
> > > **Q1**: What is the main difference of your benchmark with the previous one? Is it with symbolic language? And what do you mean by pure non-monotonic and why the previous benchmarks are not pure in the introduction?
> >
> > **Response (Q1)**:
> >
> > (1): Difference between our proposed benchmark and previous related works.
> >
> > **The proposed dataset differs from existing ones in several ways, as summarized in Table Supplement.1**.
> >
> > Existing work such as δ-NLI [a] **does not focus on non-monotonic logic reasoning** but rather on textual reasoning with non-monotonic situations; ProofWriter [b], ruletaker [c], and generics-exemplars [e] **do not contain non-monotonic reasoning scenarios**. In fact, generics-exemplars [e] only provides generic rules and exceptions (e.g., "Birds can fly, but penguins can't fly"), **without actual reasoning scenarios**. LogicNMR [d] and LogicBench [f] involve non-monotonic reasoning, but their logical structures **are relatively simple**, supporting only Default Negation and Strong Negation + Default Negation, respectively.
> >
> > (2): Explanation for pure non-monotonic reasoning.
> >
> > Pure non-monotonic reasoning means inferring **only depends on** the given facts and rules, which is the common setting in traditional non-monotonic reasoning. For example, we can not gain bird can fly if "bird(A) → canFly(A)" is not given in the rules.
> >
> > > **Q2**: At stage 2 in your method, does structure modification refer to modifying the rules template generated at stage 1 and predicate modification refers to the facts from stage 1 as well?
> >
> > **Response (Q2)**:
> >
> > No. In Stage 2, the samples are generated by modifying the structure of the Stage 1 samples, not by directly modifying the rules or predicates from Stage 1.

---

> > > ### Author Response · Authors · 2024-11-20
> > > **Response - Part 3**
> > >
> > > ## Experiments
> > >
> > > Table Supplement.1: The difference between SymTex. "Commonsense- driven" means if commonsense knowledge is required for reasoning tasks in the dataset. In the **operations** column, abbreviations indicate supported logical operations: **SN (Strong Negation)**, **DN (Default Negation)**, **Disj (Disjunction)**, and **Cons (Constraint)**. Predicate style specifies the format of predicates within the dataset: **RamS (Random String)**, **RamW (Random Word)**, **RelW (Related Word)**. The **Multi-ary predicate** and **Multi-objects** columns indicate the maximum number of arguments and objects involved in reasoning tasks, respectively. If not specified, the value is marked as N/A.
> > > | **Dataset**               | **Multi-ary predicate** | **Multi-objects** | **Commonsense- driven** | **Fact-rule-query** | **Non-monotonic** | **Operations**     | **Logic Style**  | **Predicate Style** |      |
> > > | ------------------------- | -------------------------- | ----------------- | ----------------------- | ------------------- | ----------------- | ------------------ | ---------------- | ------------------- | ---- |
> > > | **δ-NLI [a]**             |                            |                   | √                       | √                   | √                 |                    | textual          | RelW                |      |
> > > | **ProofWriter[b]**        | 2                          | 2                 | ×                       | √                   | ×                 | SN                 | textual          | RamW                |      |
> > > | **ruletaker[c]**          |                            |                   | ×                       | √                   | ×                 | SN                 | textual          | RamW                |      |
> > > | **LogicNMR[d]**           | 1                          | 1                 | ×                       | √                   | √                 | SN,DN              | textual          | RamW                |      |
> > > | **generics-exemplars[e]** |                            |                   | √                       | ×                   | ×                 |                    | textual          | RelW                |      |
> > > | **LogicBench[f]**         | 2                          | 2                 | ×                       | √                   | √                 | SN,DN              | textual          | RelW                |      |
> > > | **SymTex(ours)**          | any                        | any               | ×                       | √                   | √                 | SN, DN, Disj, Cons | textual+symbolic | RamW, RelW, RamS    |      |
> > >
> > > Table Supplement.2: This table shows the percentage of correct/incorrect prediction combinations for different models. "S" and "T" mean in the symbolic and textual setting respectively. "C" and "I" mean correct and incorrect respectively.
> > >
> > > |                       | **S:C_T:C** | **S:C_T:I** | **S:I_T:C** | **S:I_T:I** |
> > > | --------------------- | ----------- | ----------- | ----------- | ----------- |
> > > | **claude_3_5_sonnet** | 64.5        | 10.8        | 18.4        | 6.4         |
> > > | **gpt_4o**            | 44.4        | 10.2        | 30.1        | 15.3        |
> > > | **claude_3_haiku**    | 30.9        | 9.6         | 25.0        | 34.6        |
> > > | **gpt_4o_mini**       | 24.3        | 13.0        | 32.9        | 29.9        |
> > >
> > > ## References
> > >
> > > [a] R. Rudinger, V. Shwartz, J. D. Hwang, and et al. Thinking like a skeptic: Defeasible inference in natural language. In Proc. of Findings of ACL’2020, pp. 4661–4675, 2020.
> > >
> > > [b] O. Tafjord, B. D. Mishra, & Clark, P. ProofWriter: Generating implications, proofs, and abductive statements over natural language. In Proc. of Findings of ACL/IJCNLP'2021, pp. 3621-3634, 2021.
> > >
> > > [c] P. Clark, O. Tafjord, & K. Richardson. Transformers as soft reasoners over language. In Proc. of IJCAI'2021, pp. 3882-3890, 2021.
> > >
> > > [d] Y. Xiu, Z. Xiao, and Y. Liu. Logicnmr: Probing the non-monotonic reasoning ability of pre-trained language models. In Proc. of Findings of EMNLP’2022, pp. 3616–3626, 2022.
> > >
> > > [e] E. Allaway, J. D. Hwang, C. Bhagavatula, and et al. Penguins don’t fly: Reasoning about generics through instantiations and exceptions. In Proc. of EACL’2023, pp. 2610–2627, 2023.
> > >
> > > [f] M. Parmar, N. Patel & N. Varshney. LogicBench: Towards systematic evaluation of logical reasoning ability of large language models. In Proc. of ACL'2024, pp. 13679-13707, 2024.
> > >
> > > [1] Y. Wang, H. Le, A. Gotmare, and et al. CodeT5+: Open Code Large Language Models for Code Understanding and Generation. In Proc. of EMNLP'2023, pp. 1069-1088, 2023.
> > >
> > > [2] D. Nam, A. Macvean, V. Hellendoorn, and et al. Using an llm to help with code understanding. In Proc. of the IEEE/ACM 46th International Conference on Software Engineering, pp. 1-13, 2024.
> > >
> > > [3] J. McCarthy. Circumscription—a form of non-monotonic reasoning. Artif. Intell., 13(1-2):27–39, 1980.
> > >
> > > [4] R. Reiter. A logic for default reasoning. Artif. Intell., 13(1-2):81–132, 1980.

---

> > > > ### Comment · Reviewer_9xu5 · 2024-11-22
> > > >
> > > > Thank you for your detailed response. Most of my questions have been addressed, and I will raise my rating accordingly.
> > > >
> > > > Your clarification of the motivation, methodology, and the purpose of each subset is much clearer now. I encourage you to update those sections correspondingly in your revision to reflect these improvements.
> > > >
> > > > That said, I feel my question regarding the methodology has not been fully addressed. Specifically, when I asked, "There are various types of symbolic rules; how do you determine which specific rule to use?", I was not referring to the symbolic syntax or merely confirming that you are using ASP syntax. This was already clear to me. My question is about the specific ASP rules you are employing and the rationale behind generating them.
> > > >
> > > > Specifically, in Figure 2, Stage 1: Generation, you show five different rules. Are these the only rule templates included in your approach? Please give a clarification regarding the coverage of the rules. For example, first-order logic comes with a set of predefined rules. So what does your set of generated ASP rules cover?

---

> > > > > ### Author Response · Authors · 2024-11-23
> > > > >
> > > > > Thank you for taking the time to review and respond to our comments. We truly appreciate your feedback and provide clarifications here.
> > > > >
> > > > > > add. Q1: Are these the only rule templates included in your approach?
> > > > >
> > > > > **Answer (add. Q1):**
> > > > >
> > > > > For templates in Stage 1: Generation.
> > > > >
> > > > > **No**. Figure 2 only shows one simple template to display the template generation process. In practice, we generate various templates by adjusting the parameters, which are listed in Table 2. Additionally, Figures 5, 6, and 7 show several final samples from different templates.
> > > > >
> > > > > > add. Q2: So what does your set of generated ASP rules cover?
> > > > >
> > > > > **Answer (add. Q2):**
> > > > >
> > > > > For rule coverage.
> > > > >
> > > > > In our generation framework, templates in Stage 1 (Generation) initially include only positive programs, while special construct, such as  "Negation as Failure" and "Disjunctive Rules", are introduced in Stage 2 (Modification).
> > > > >
> > > > > We have summarized and listed the ASP's constructs in Table Supplement.4. **Our dataset covers most of the constructs of ASP programs and we support all the core features of ASP ("Negation as Failure" and "Disjunctive Rules"). Note that the constructs we do not support all belong to ASP extensions extensions or syntax sugar.**
> > > > >
> > > > >
> > > > >
> > > > > ## Experiments
> > > > >
> > > > > Table Supplement.4 Common constructs of ASP programs.
> > > > >
> > > > > | **Construct**               | **Explanation**                                              | **Example**                                             | **SymTex** |
> > > > > | --------------------------- | ------------------------------------------------------------ | ------------------------------------------------------- | ---------- |
> > > > > | **Atoms**                   | Basic facts or entities in the domain.                       | bird(sparrow)                                           | √          |
> > > > > | **Literals**                | An atom or its negation.                                     | fly(sparrow) or - fly(sparrow)                          | √          |
> > > > > | **Rules**                   | Implications that define relationships between atoms (head :- body). | fly(X) :- bird(X), - penguin(X).                        | √          |
> > > > > | **Facts**                   | Ground rules with no body, representing axioms.              | bird(sparrow).                                          | √          |
> > > > > | **Constraints**             | Rules without heads, used to restrict valid solutions.       | :- fly(X), penguin(X).                                  | √          |
> > > > > | **Choice Rules**            | Rules defining optional inclusion of atoms in answer sets.   | {fly(X)} :- bird(X).                                    |            |
> > > > > | **Cardinality Constraints** | Bounds on the number of satisfied literals.                  | 1 { fly(X) : bird(X) } 2.                               |            |
> > > > > | **Aggregates**              | Functions (sum, count, min, max) applied to collections of literals. | totalWeight(W) :- W = #sum { weight(X) : selected(X) }. |            |
> > > > > | **Negation as Failure**     | True if a literal cannot be proven true (negation by failure). | safe(X) :- not unsafe(X).                               | √          |
> > > > > | **Strong Negation**         | Classical negation, explicitly denoted by -.                 | -fly(X) :- penguin(X).                                  | √          |
> > > > > | **Disjunctive Rules**       | Rules with multiple possible outcomes (disjunction in the head). | fly(X) \| swim(X) :- bird(X).                           | √          |
> > > > > | **Optimization Statements** | Used to minimize or maximize an objective function.          | #minimize { cost(X): selected(X) }.                     |            |

---

> > > > > > ### Comment · Reviewer_9xu5 · 2024-11-25
> > > > > >
> > > > > > Thank you for providing the supplementary table; it perfectly addresses what I was looking for. I recommend including this table in your revision (e.g., in the appendix) as it will help readers better understand the rule constructs. Accordingly, I have raised the rating of your paper.

---

> > > > > > > ### Author Response · Authors · 2024-11-27
> > > > > > > **The revised paper has been submitted.**
> > > > > > >
> > > > > > > > Revised Paper
> > > > > > >
> > > > > > > **The revised paper has been submitted.**
> > > > > > >
> > > > > > > We sincerely appreciate the valuable feedback provided by the reviewers and have submitted the revised paper accordingly. We hope the revisions could address the reviewers' concerns, and we welcome any further questions or suggestions that may arise.

---

### Official Review · Reviewer_rb9W · 2024-11-03

**Soundness:** 3
**Presentation:** 3
**Contribution:** 2
**Rating:** 6
**Confidence:** 3

**Summary:**

A new dataset to evaluate the reasoning powers of LLMs is proposed and several state-of-the-art LLMs are evaluated on this dataset. The main purpose is to evaluate whether an LLM can perform non-monotonic reasoning. The framework is based on Answer Set Programming and a symbolic program is generated and then mapped to a textual description. The tasks evaluated are computing a boolean query and computing an answer set for the program. Several state-of-the-art LLMs are evaluated on the generated dataset and conclusions are presented on their performance in the 2 tasks.

**Strengths:**

Strengths
- A new dataset based on ASP seems like a novel contribution and can have significance and impact in the Neuro-symbolic community
- The experiments seem well thought out and include several LLMs and helps us evaluate logical reasoning in LLMs
- Sharing the datasets will benefit the community

**Weaknesses:**

Weaknesses
- As far as I understood, existing work (and mentioned in the related work) has focussed on more text-based non-monotonic reasoning capabilities and the main contribution of the proposed dataset is to evaluate if the LLMs can process symbolic representations. What were the results from the earlier work, how does this new work add/modify our views of LLM capabilities? In general, I felt like placing the new dataset in the context of what is known about LLMs using existing work and what new knowledge the dataset and results are adding is important in an empirical paper such as this one.
- In general, why would it make sense for an LLM (built primarily for language) to be evaluated with a symbolic program was not something that was very clear. Related to this, is there a real-world use-case that motivates this usage of LLMs? (There are a couple of sentences mentioned but a more clear motivation would be useful)
- One of the steps in the proposed approach also performs textualization of the program, how sensitive is the LLM to the quality of going from symbolic models to an equivalent text representation?
- Regarding the number of instances, it is mentioned 1000 instances were used (in the experiments) but the dataset is much larger (178K) , is there a reason for this disparity or is there some additional explanations needed here to replicate the results?

**Questions:**

- Some more detail about the benefit of the proposed method (in relation to the types of evaluation already conducted in a similar domain) would be helpful. (See weaknesses)

---

> ### Author Response · Authors · 2024-11-20
> **Response - Part 1**
>
> We greatly appreciate the time and effort you have invested. In response to your concerns, we have provided clarifications here.
>
> > **W1**:As far as I understood, existing work (and mentioned in the related work) has focussed on more text-based non-monotonic reasoning capabilities and the main contribution of the proposed dataset is to evaluate if the LLMs can process symbolic representations. What were the results from the earlier work, how does this new work add/modify our views of LLM capabilities? In general, I felt like placing the new dataset in the context of what is known about LLMs using existing work and what new knowledge the dataset and results are adding is important in an empirical paper such as this one.
>
> **Response (W1):**
>
> (1): The results from earlier works.
>
> Earlier studies have demonstrated that LLMs face challenges when dealing with non-monotonic reasoning problems. **However, the samples in these works suffer from relatively simple construction or driven by common-sense knowledge, which can not fully or purely evaluate LLMs' capability on non-monotonic reasoning.**
>
> (2): New views from our benchmark.
>
> From the results in Table Supplement.2, we can suggest that there is **a complement between symbolic and textual samples.**
>
> > **W2**: In general, why would it make sense for an LLM (built primarily for language) to be evaluated with a symbolic program was not something that was very clear. Related to this, is there a real-world use-case that motivates this usage of LLMs? (There are a couple of sentences mentioned but a more clear motivation would be useful)
>
> **Response (W2):**
>
> (1): Reason to evaluate the symbolic programs on LLMs.
>
> The logic program, such as the ASP programs in our benchmark, is also a type of code, which is highly related to code generation or code understanding tasks on LLMs\[1\]\[2\], and non-monotonic reasoning has a well-established and mature symbolic foundation \[3\]\[4\]. **The LLMs' capability of understanding and solving the logic programs can represent their ability for logic reasoning and instruction following**.
>
> (2) The usage of symbolic programs.
>
> Table Supplement.2 suggests that **if both formats are used together for reasoning, LLMs can potentially produce more accurate answers. This also underscores the necessity of symbolic logic samples.**
>
> > **W3**: One of the steps in the proposed approach also performs textualization of the program, how sensitive is the LLM to the quality of going from symbolic models to an equivalent text representation?
>
> **Response (W3)**:
>
> (1): For data quality.
>
> This work focuses on scenarios that the non-monotonic reasoning logic programs can describe, especially by answer set programming (ASP). In this case, **we can evaluate the correctness of generated samples by executing the programs on ASP solvers.**
>
>  (2): For equivalent between symbolic and textual samples.
>
> In practice, textual samples are often converted back into symbolic samples. For example, both "X eats Y" and "X and Y are friends" can be represented symbolically as a(X, Y). To ensure equivalence between symbolic and textual samples, **we employ a template-based approach that facilitates a consistent transformation between the two, preserving their logical integrity.**

---

> > ### Author Response · Authors · 2024-11-20
> > **Response - Part 2**
> >
> > > **W4**: Regarding the number of instances, it is mentioned 1000 instances were used (in the experiments) but the dataset is much larger (178K) , is there a reason for this disparity or is there some additional explanations needed here to replicate the results?
> >
> >
> >
> > **Response (W4)**:
> >
> > The reason for using 1000 instances in the experiments was mainly due to time constraints. Furthermore, we have **saved the selected instances** to ensure that the results can be replicated in future work.
> >
> > ### Experiments
> >
> > Table Supplement.2: This table shows the percentage of correct/incorrect prediction combinations for different models. "S" and "T" mean in the symbolic and textual setting respectively. "C" and "I" mean correct and incorrect respectively.
> >
> > |                       | **S:C_T:C** | **S:C_T:I** | **S:I_T:C** | **S:I_T:I** |
> > | --------------------- | ----------- | ----------- | ----------- | ----------- |
> > | **claude_3_5_sonnet** | 64.5        | 10.8        | 18.4        | 6.4         |
> > | **gpt_4o**            | 44.4        | 10.2        | 30.1        | 15.3        |
> > | **claude_3_haiku**    | 30.9        | 9.6         | 25.0        | 34.6        |
> > | **gpt_4o_mini**       | 24.3        | 13.0        | 32.9        | 29.9        |
> >
> >
> >
> > ### Reference
> >
> > [1] Y. Wang, H. Le, A. Gotmare, and et al. CodeT5+: Open Code Large Language Models for Code Understanding and Generation. In Proc. of EMNLP'2023, pp. 1069-1088, 2023.
> >
> > [2] D. Nam, A. Macvean, V. Hellendoorn, and et al. Using an llm to help with code understanding. In Proc. of the IEEE/ACM 46th International Conference on Software Engineering, pp. 1-13, 2024.
> >
> > [3] J. McCarthy. Circumscription—a form of non-monotonic reasoning. Artif. Intell., 13(1-2):27–39, 1980.
> >
> > [4] R. Reiter. A logic for default reasoning. Artif. Intell., 13(1-2):81–132, 1980.

---

> > > ### Comment · Reviewer_rb9W · 2024-11-25
> > >
> > > Thanks for the response, it answers many of my questions. Regarding the time constraints, maybe I missed how long it takes to evaluate using the dataset. I am guessing the bottleneck is the usage of LLMs (and not eh MG-Symtex pipeline as such). This means is it that we can generate large datasets but it is  hard to evaluate them at scale on LLMs? The reason I ask is, would the type of sample you choose bias the results? Given that the main contribution of the paper is to introduce a new (fairly large) dataset to analyze the capabilities of LLMs, I am wondering if scaling up evaluation (or practical use of the dataset) will be an issue.

---

> > > > ### Author Response · Authors · 2024-11-27
> > > >
> > > > Thanks for your valuable feedback. In response to your concerns, we have provided clarifications here.
> > > >
> > > > Regarding the time constraints, as you mentioned, the bottleneck is the usage of LLMs rather than the MG-Symtex itself.
> > > >
> > > > Regarding your concern, the choice of samples could potentially introduce bias into the results, as different types of samples might affect model performance differently. **To address this, we conduct experiments in Section 5.3 in the revised paper, titled "Variable Impact Analysis on Results"**, where we analyze the effects of varying query arity, related facts and rules, as well as noisy facts and rules on F1 scores.
> > > >
> > > > > Revised Paper
> > > >
> > > > **The revised paper has been submitted.**
> > > >
> > > > We sincerely appreciate the valuable feedback provided by the reviewers and have submitted the revised paper accordingly. We hope the revisions could address the reviewers' concerns, and we welcome any further questions or suggestions that may arise.

---

> > > > ### Author Response · Authors · 2024-11-29
> > > >
> > > > Dear reviewer rb9W,
> > > > ﻿
> > > >
> > > > Your suggestions have been incredibly helpful in improving our paper. The content of the rebuttal will be incorporated into the paper. If you have any further questions, please feel free to ask. Your recognition is important to us. Thank you very much!

---

### Official Review · Reviewer_SK7G · 2024-11-04

**Soundness:** 3
**Presentation:** 4
**Contribution:** 2
**Rating:** 5
**Confidence:** 3

**Summary:**

The paper introduces SymTex, a benchmark designed to evaluate large language models' (LLMs) non-monotonic reasoning (NMR) capabilities, where conclusions may be retracted upon receiving new information. SymTex includes both symbolic and textual representations to facilitate an analysis of LLMs' reasoning abilities across these forms. The authors propose a novel data generation framework, MG-SymTex, that generates diverse non-monotonic samples in symbolic and textual formats, enabling a balanced evaluation across forms. Two main tasks are defined for the benchmark: Tri-State Boolean Querying and Answer Set Computation. Comprehensive experiments show that LLMs face substantial challenges with non-monotonic reasoning tasks, particularly in symbolic settings.

**Strengths:**

1. well-written and good presentation.
2. interesting benchmark for a "challenging" task: The paper focuses on non-monotonic reasoning. By setting up a dual-format dataset (symbolic and textual), the benchmark provides a balanced framework that probes the models' adaptability in response to new information, an essential aspect of human-like reasoning.
3. sound methodology: The generation process (MG-SymTex) is well-structured, detailing how symbolic samples are transformed into text, ensuring consistency between both formats. This methodological rigor allows for a controlled comparison of LLM performance on both textual and symbolic reasoning tasks, making the dataset a reliable evaluation tool.
4. comprehensive evaluation: The authors evaluate eight LLMs across various reasoning tasks in SymTex, providing detailed analyses and metrics. This analysis reveals the strengths and limitations of models in handling symbolic logic compared to textual logic, offering a clearer understanding of their reasoning capabilities.

**Weaknesses:**

1. The authors note that LLMs perform better on textual representations than on symbolic ones, but the paper lacks specific solutions or insights into how to address this gap. An in-depth exploration of architectural or methodological changes that could help LLMs better handle symbolic reasoning would add value.
2. One area for potential improvement in the benchmark's symbolic setting is the integration of reasoning interpreters (e.g., ASP solvers) alongside LLMs. While evaluating LLMs independently provides an insightful baseline of their inherent reasoning capabilities, it might also be valuable to explore how these models perform in combination with interpreters. This would reflect a practical, real-world scenario where LLMs often rely on auxiliary tools for enhanced reasoning.
3. Potential Dataset Bias: As with any generated dataset, SymTex may contain unintentional biases due to the template-based generation process. Without rigorous diversity checks, the dataset might fail to capture the full spectrum of possible reasoning challenges, leading models to overfit specific structures. Providing a bias analysis or diversity evaluation for the generated samples would help ensure a robust assessment of LLMs’ reasoning abilities.

**Questions:**

N/A

---

> ### Author Response · Authors · 2024-11-20
>
> We greatly appreciate the time and effort you have invested. In response to your concerns, we have provided clarifications here.
>
>
>
> > **W1**:The authors note that LLMs perform better on textual representations than on symbolic ones, but the paper lacks specific solutions or insights into how to address this gap. An in-depth exploration of architectural or methodological changes that could help LLMs better handle symbolic reasoning would add value.
>
> **Response (W1):** In Section 6 ('Conclusion'), we provide some preliminary insights and suggests future directions, including potential approaches to bridge the gap between symbolic reasoning and the capabilities of LLMs.
>
>
>
> > **W2**: One area for potential improvement in the benchmark's symbolic setting is the integration of reasoning interpreters (e.g., ASP solvers) alongside LLMs. While evaluating LLMs independently provides an insightful baseline of their inherent reasoning capabilities, it might also be valuable to explore how these models perform in combination with interpreters. This would reflect a practical, real-world scenario where LLMs often rely on auxiliary tools for enhanced reasoning.
>
> **Response (W2)**: We agree that combining reasoning tools like ASP solvers with LLMs could be a valuable direction. This aligns with our future research plans, and we're already exploring how such collaboration can improve the reasoning abilities of both models.
>
>
>
> > **W3**: Potential Dataset Bias: As with any generated dataset, SymTex may contain unintentional biases due to the template-based generation process. Without rigorous diversity checks, the dataset might fail to capture the full spectrum of possible reasoning challenges, leading models to overfit specific structures. Providing a bias analysis or diversity evaluation for the generated samples would help ensure a robust assessment of LLMs’ reasoning abilities.
>
> **Response (W3)**:
>
> (1): Diversity of samples.
>
> To ensure diverse sample generation, we adjust various parameters, such as the number of facts and rules, as well as the maximum arity of rules and predicates.
>
> (2): Multi-run results.
>
> Biases in logical program datasets can be difficult to define and identify. To mitigate this issue, we performed three independent runs for each experiment and reported the average results.

---

> > ### Comment · Reviewer_SK7G · 2024-11-25
> >
> > Thank you for your response. I’ll maintain my score.

---

> > > ### Author Response · Authors · 2024-11-29
> > >
> > > Dear reviewer SK7G,
> > > ﻿
> > >
> > > Your suggestions have been incredibly helpful in improving our paper. The content of the rebuttal will be incorporated into the paper. If you have any further questions, please feel free to ask. Your recognition is important to us. Thank you very much!

---

> ### Author Response · Authors · 2024-11-27
> **The revised paper has been submitted.**
>
> > For dataset bias
>
> Regarding your concern, SymTex may contain unintentional biases, as different types of samples might affect model performance differently. **To address this, we conduct experiments in Section 5.3 in the revised paper, titled "Variable Impact Analysis on Results"**, where we analyze the effects of varying query arity, related facts and rules, as well as noisy facts and rules on F1 scores.
>
> > Revised Paper
>
> **The revised paper has been submitted.**
>
> We sincerely appreciate the valuable feedback provided by the reviewers and have submitted the revised paper accordingly. We hope the revisions could address the reviewers' concerns, and we welcome any further questions or suggestions that may arise.

---

### Meta-Review · Area_Chair_vBcA · 2024-12-11

**Metareview:**

In this work, the authors introduce a framework called Multi-step Generation for Symbolic and Textual NMR Samples (MG-SymTex) to generate diverse non-monotonic samples automatically, and build a non-monotonic reasoning benchmark, called SymTex, which is used to evaluate the non-monotonic reasoning capability of LLMs. The proposed SymTex comprises two types of description and three types of predicate, facilitating two primary tasks including Tri-State Boolean Querying and Answer Set Computation. Through comprehensive evaluations, the authors demonstrate that state-of-the-art LLMs such as gpt-4o, claude-3.5-sonnet, and o1-mini encounter significant challenges when addressing the proposed benchmark, highlighting the difficulty of non-monotonic reasoning in LLMs.

This is a boarderline paper, and some of the reviewers doubt about the direct evidence to justify the necessity of an non-monotonic ASP benchmark for LLMs. Specifically, the reviewer would expect either (1) real-world use cases such as code generation that exactly match the form of ASP programs in the proposed benchmark; or (2) evidence showing that existing LLMs perform better on code generation if they perform better on SymTex.

**Additional Comments On Reviewer Discussion:**

The reviewers and authors have been actively involved in the rebuttal stage, and the review committe has considered the rebuttal in a very careful way. In general, they consider that this paper still contains some unclear issues, therefore, due to the limited positions, I think this paper is immature to be accepted in its current version.

---

### Decision · Program_Chairs · 2025-01-22

Reject